# HOW TO TEACH LARGE MULTIMODAL MODELS NEW SKILLS?

## ABSTRACT

How can we teach large multimodal models (LMMs) new skills without erasing prior abilities? We study sequential fine-tuning on five target skills while monitoring general ability on eight held-out benchmarks across three model families. We observe that apparent "forgetting" on held-out tasks after narrow fine-tuning can partly recover at later stages. We trace this behavior to a measurable shift in the output token distribution, manifested through a simple counting-bias probe that identifies the shift co-varies with forgetting. Guided by this picture, we identify two simple, robust tuning recipes that learn strongly while limiting drift: (i) updating only the self-attention projection layers, and (ii) updating only the MLP Gate&Up while freezing the Down projection. Across models and tasks, these choices deliver strong target gains while largely preserving held-out performance.

## 1 INTRODUCTION

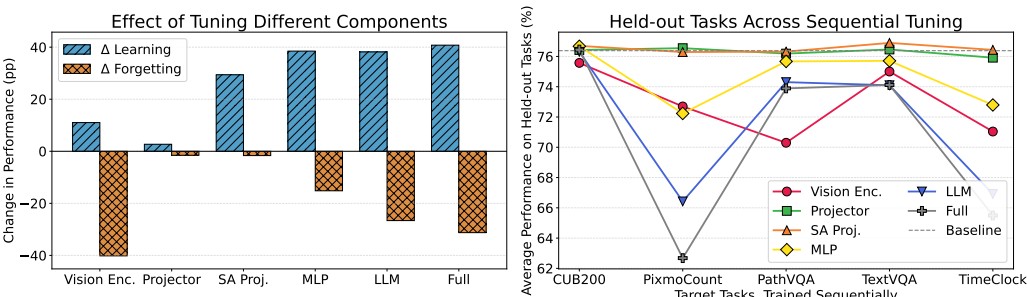

Figure 1: **Surprising Forgetting Behavior in LMMs**: **Left**: When fine-tuning most components on one target task, we see major improvement in that task ("Learning") but a substantial drop in performance of other tasks ("Forgetting", total across tasks shown here), as expected. But if we only tune self-attention projection layers (SA Proj.) in the language model, we still get substantial learning on the target task with minimal forgetting. **Right**: Even fine-tuning SA Proj. for multiple tasks sequentially, we see no forgetting. For others, we see large forgetting on the PixmoCount task, but the models somehow partly recover what they "forgot" in learning the next specialized task. Our paper documents and analyzes these and other interesting phenomena of learning and forgetting in LMMs, leading to simple and effective ways to teach LMMs new skills.

Large multimodal models (LMMs), such as LLaVA (Liu et al., 2023b) and Qwen2.5-VL (Bai et al., 2025), are trained to generate natural language answers based on image(s) and natural language instruction. As such, these models can perform a wide range of tasks. However, for many special domains, such as medical images, or skills, such as counting, the models do not perform as desired. How can we teach LMMs something new without degrading existing capabilities?

Training a new LMM can cost millions of dollars, weeks of time, and emit hundreds of tons of $CO_2$, so finding ways to more efficiently and effectively update existing models is a pressing concern.

One option is to simply fine-tune the model on the new task. However, at least for simpler models, fine-tuning is known to cause catastrophic forgetting, such that a model previously proficient on many tasks becomes a narrow expert on the new one. A more reliable method is to completely retrain the model with an expanded training set, but this becomes increasingly impractical as the scale of training

data continues to climb. Intuitively, an intelligent system should be able to add to its knowledge without repeating all of its learning. LMMs are sometimes trained in a single epoch (Li et al., 2024c) raising a pressing question: do LMMs suffer catastrophic forgetting? Recent works (Chen et al., 2024a; Yu et al., 2024; Zhu et al., 2024a) conclude yes, but our findings are more nuanced.

We study continual learning in LMMs using a controlled evaluation program. The target suite contains five practical skills that span different answer formats (fine-grained bird classification, counting, medical VQA, OCR reading, and time reading). The held-out suite contains eight widely used benchmarks for general vision–language ability. We evaluate *learning* as improvement on the target tasks and *forgetting* as the average drop on held-out tasks.

Our first goal is to identify tunable parts that deliver high target performance with minimal forgetting. We compare full-model fine-tuning to tuning each major component (vision encoder, projector, LLM) and then open the LLM into its two essential blocks—self-attention projections (SA Proj.) and the feed-forward network (MLP). Early experiments on LLaVA-OneVision (Fig. 1) reveal two surprising results: 1) tuning SA Proj. learns with little or no measurable forgetting across a five-task sequence; and 2) what appears forgotten after one stage can be recovered by tuning another specialized task.

These results lead us to ponder: why is SA Proj. so robust to forgetting, and how is forgotten knowledge recovered without rehearsing? Consider the roles of the two essential components in the transformer decoder: self-attention projection is data processing, applying an algorithm to the inputs (Elhage et al., 2021; Olsson et al., 2022), while MLPs perform external memory look up and produce the output distribution (Geva et al., 2021). We thus hypothesize that perhaps what looks like forgetting or interference after fine-tuning on a narrow target task is actually bias in the output distribution due to the task distribution shift. Through in-depth analysis when tuning the counting task, we confirm this hypothesis: tuning the MLP increases target accuracy but also increases the likelihood of outputting numeric tokens and a highly correlated drop in held-out task accuracy, while tuning the self-attention achieves the target learning without much bias toward numeric tokens and without losing held-out accuracy (Sec. 5.2).

Guided by this result, we explore tuning recipes that preserve learning while limiting output shift. To avoid biasing the output distribution, we tune the MLP up/gating projections while keeping the down projection frozen, and find that it achieves similar learning to full MLP tuning with little forgetting. We experiment on LLaVA-OneVision (Li et al., 2024c) by training five target tasks sequentially, averaging over three sequence orders, measuring the learning and forgetting in target tasks and held-out tasks (Sec. 5.1). We then confirm that similar trends hold for LLaVA-NeXT (Li et al., 2024b) and Qwen2.5-VL (Bai et al., 2025) (Sec. 5.3).

In summary, our work documents and analyzes several interesting phenomena of learning and forgetting in LMMs, leading to simple and effective ways to teach LMMs new tricks. The findings are:

- **Tuning the LLM** ($\triangle$ learning +31.8/$\triangle$ forgetting -23.3) **is critical for learning new tasks**, while tuning the vision encoder (+9.6/-10.8) brings little gain and harms general ability.

- **Tuning only the self-attention projection weights (+24.9/-0.6) or the up layers of the MLP (+30.5/-2.1) provides excellent learning with limited forgetting**, evaluated on a five-target task sequence, eight held-out benchmarks, and three model families.

- **Forgetting is largely a manifestation of output distribution shift.** We use a simple counting-bias probe to show that the rise in number-token likelihood grows with MLP tuning and remains near baseline for self-attention tuning; the magnitude of this shift co-varies with held-out drops. Therefore, forgetting can be recovered when subsequent tuning shifts back the output distribution, and methods that limit shift effectively mitigate forgetting, such as distillation to the previous checkpoint or freezing the MLP down projection while tuning the up&gate.

## 2 RELATED WORK

**Continual learning for traditional vision.** Continual learning, also known as lifelong learning (Aljundi et al., 2017; Chen & Liu, 2018; Chaudhry et al., 2019), aims to train models on a sequence of tasks or data streams without forgetting previously acquired knowledge. Traditionally, it is mainly explored in closed-vocabulary image classification, and can be categorized into three main

types: (1) *regularization-based* methods try to preserve the knowledge captured in a previous version of the model by matching logits (Li & Hoiem, 2017; Rebuffi et al., 2017), feature maps (Douillard et al., 2020), or other information (Tao et al., 2020; Wang et al., 2022; Simon et al., 2021; Joseph et al., 2022; PourKeshavarzi et al., 2022; Liu et al., 2023c) in the new model; (2) *exemplar replay* methods build a reservoir of samples from old training rounds (Prabhu et al., 2020; Liu et al., 2024b; Luo et al., 2023b; Liu et al., 2020; Rebuffi et al., 2017; Shin et al., 2017; Bang et al., 2021) and replay them in successive training phases as a way of recalling past knowledge; and (3) *network-architecture-based* methods (Liu et al., 2021; Wang et al., 2022) expand the network capacity for new target data and freeze some network parameters to retain original knowledge. Recently, several studies (Jin et al., 2022; Khattak et al., 2023a;b; Smith et al., 2023) show prompt tuning as an effective strategy for continual learning, which freezes all weights but adds learnable prompts to optimize for new tasks.

**Vision-text contrastive models**, such as CLIP (Radford et al., 2021), are trained to align images and texts for open-vocabulary image classification and retrieval. Pretrained CLIP models may outperform on fine-grained or specialized tasks (Radford et al., 2021; Zhu et al., 2024c), pressing the need for reliable continual learning approaches. Zhu et al. (2024c;b) propose learning to blend predictions from original and tuned image encoders, enabling fast online learning without forgetting for open vocabulary classification. Yu et al. (2024) adds parameter-efficient adapters to a mixture-of-experts on a frozen CLIP model to prevent forgetting. Zhou et al. (2025) design task-specific projection layers and cross-modal fusion modules for vision-language models in class-incremental learning. Liu et al. (2025) incorporate continual low-rank adaptation and knowledge consolidation to prevent forgetting. Zheng et al. (2023) use knowledge distillation (Li & Hoiem, 2017) on CLIP to maintain zero-shot performance.

**Large language models (LLMs).** Studies of LLMs evaluate the learning-forgetting trade-off across various fine-tuning strategies on LLMs with billions of parameters, including full-model tuning, adapters, LoRA, and prompt tuning. Luo et al. (2023a) find that decoder-only models are more robust than encoder-decoder models. Lin et al. (2024) observe that models fine-tuned for narrow domains lose ability on general tasks, but method like weight interpolation (WiSE-FT) (Wortsman et al., 2022) helps maintain balance. Biderman et al. (2024) show that LoRA reduces learning and forgetting, compared to full fine-tuning. Li et al. (2025) propose a dual-memory replay framework with interpolated LoRA. Huang et al. (2024) generate pseudo-data from the model itself to mitigate forgetting without requiring original training data. Xiang et al. (2023) propose to use regularization strategies such as Elastic Weight Consolidation (EWC) (Kirkpatrick et al., 2017) and hierarchical importance-based penalties to preserve general knowledge by constraining updates to important parameters. Wang et al. (2023) propose learning orthogonal LoRA weights for new tasks to mitigate forgetting.

**Roles of attention and FFN in LLMs.** Mechanistic studies of transformer blocks show a division of labor. Attention heads act primarily as *routing* and retrieval mechanisms: they select *where* to read from using query–key patterns and then mix the corresponding values; this view is formalized in the Transformer Circuits framework and supported by analyses of "induction heads," which implement a simple copying algorithm and closely track the emergence of in-context learning during training (Olsson et al., 2022). In contrast, feed-forward (FFN/MLP) blocks behave like *key–value memories*: learned keys detect input patterns while values write features that align with groups of vocabulary items, thereby shifting the model's output preferences (Geva et al., 2021). Meng et al. (2022) show that directly modifying MLP weights updates specific facts while preserving unrelated behavior, implying FFN as a principal site where "what to say" is stored. Earlier analyses in BERT and NMT also found that a minority of attention heads specialize into linguistically interpretable roles (e.g., syntax, coreference) while many heads are prunable with little loss, reinforcing the view of attention as selective routing rather than the main repository of lexical knowledge (Clark et al., 2019; Voita et al., 2019). This literature aligns with our empirical finding that self-attention updates tend to preserve global behavior while MLP updates are the main driver of output-distribution shift.

**Large multimodal models (LMMs).** Relatively little work has investigated continual learning in LMMs, but there is growing interest. Chen et al. (2024a) find that LMMs suffer from catastrophic forgetting when learning a sequence of new tasks. Most studies of LMMs more narrowly focus on visual question answering (VQA) (Zhang et al., 2023; Nikandrou et al., 2024; Lin et al., 2025; Marouf et al., 2025; Del Chiaro et al., 2020) or image captioning (Nguyen et al., 2019). Zhang et al. (2023) leverage both sample-specific and sample-invariant features to learn representations that are both discriminative and generalizable for VQA tasks. Nikandrou et al. (2024) propose to distill knowledge

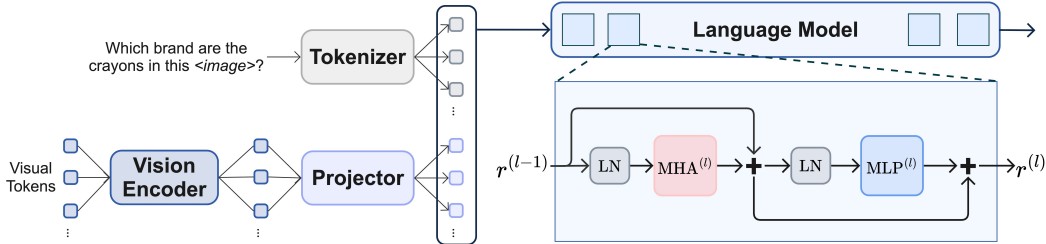

Figure 2: Architecture of our evaluated LMMs. The input contains visual inputs such as images or videos, which are converted to visual tokens by the vision encoder, and text input is processed by a tokenizer containing a visual placeholder token `<image>`. Visual tokens are converted by the projector and concatenated with text tokens as input for the language model. We visualize the architecture of the transformer decoder layer of the language model. "LN", "MHA", "MLP" represent layer norm, multi-head attention, and multi-layer perceptron, respectively. $r^{(l)}$ is the final output of layer $l$.

separately for each modality, ensuring that both image features and question features retain their relevant information when new tasks arrive. Lin et al. (2025) combine selective memory replay and knowledge distillation for VQA. Marouf et al. (2025) store only past questions from previous tasks as memory for rehearsal. For older models, Nguyen et al. (2019) integrate continual learning techniques, such as finetuning schemas and regularization, into the captioning pipeline to combat forgetting, and Del Chiaro et al. (2020) introduce an attention-based LSTM architecture.

Our work complements these studies in several ways: (1) more diverse tasks, such as counting, clock reading, classification, OCR, and medical VQA, finding large differences in the extent of learning and forgetting in each and that what is "forgotten" by one task can be recovered by learning the next (see Figs. 1 and 4); (2) systematic analysis that forgetting is highly related to output token distribution shift and methods that prevents shifts mitigate forgetting (Sec. 5.2 and 5.4); (3) investigation into tuning different components, finding that tuning MLP Gate&Up and SA Proj. provide a good balance of learning and forgetting.

## 3 METHOD

**Setting.** We adapt a pretrained large multimodal model on either a *single-target* task or a *sequential* stream of tasks. In the single-target case, given a target dataset $\mathcal{D}_T$ and a held-out suite $\mathcal{D}_H$, the goal is to improve performance on $\mathcal{D}_T$ while preserving performance on $\mathcal{D}_H$. In the sequential case, tasks $\{\mathcal{D}_T^{(1)}, \ldots, \mathcal{D}_T^{(K)}\}$ arrive in stages; unless noted we update at each stage without rehearsal (no mixing of earlier tasks) and assess both the current target and the aggregated held-out suite after every stage.

### 3.1 MODEL

**Overview of the LMM.** Our evaluated LMMs have three major parts (Fig. 2): a vision encoder that turns an image into visual tokens, a projector that maps those tokens to the language width $d$, and a decoder-only transformer language model that produces next-token logits given the visual and text tokens.

**Vision encoder and projector.** The vision encoder produces $v = f_{\text{vis}}(I) \in \mathbb{R}^{S_v \times d_v}$. The projector maps to the language representation width,

$$x_{\text{vis}} = g_\psi(v) \in \mathbb{R}^{S_v \times d},$$

where $\psi$ are the projector's trainable parameters.

**Language model.** The language model is a pre-norm, decoder-only transformer with $L$ identical blocks. As illustrated in Fig. 2, the sublayer outputs and residual update at block $l$ are

$$a^{(l)} = \text{MHA}^{(l)}\big(\text{LN}(r^{(l-1)})\big), \quad f^{(l)} = \text{MLP}^{(l)}\big(\text{LN}(r^{(l-1)} + a^{(l)})\big), \quad r^{(l)} = r^{(l-1)} + a^{(l)} + f^{(l)}, \quad (1)$$

where MHA and MLP denote the **multi-head self-attention** and **feed-forward network** and LN is layer normalization.

*Self-attention.* With input $x$ and per-head key width $d_k$,

$$\{Q, K, V\} = x\{W_Q, W_K, W_V\} \xrightarrow{\text{attention}} A = \text{softmax}\left(\frac{QK^\top}{\sqrt{d_k}}\right) \xrightarrow{\text{value mix} + W_O} \text{MHA}(x) = (AV)W_O \quad (2)$$

Here $W_Q, W_K$ set where to attend (routing), $W_V$ selects the content to mix, and $W_O$ is the matrix that writes the attention result back into the residual stream at model width $d$.

*Feed-forward.* With input $x$ and gating nonlinearity $\phi = \text{SiLU}$,

$$\text{MLP}(x) = W_{\text{down}}(\phi(xW_{\text{gate}}) \odot (xW_{\text{up}})), \quad (3)$$

where $W_{\text{gate}}, W_{\text{up}}$ detect features (key-like pattern match), and $W_{\text{down}}$ writes those features back to the residual at width $d$. We use $U \in \mathbb{R}^{d \times |V|}$ for the LM head and denote the final block output as $r^{(L)}$, so logits are $z = U^\top r^{(L)}$.

**Residual stream.** Let $x_{\text{text}} \in \mathbb{R}^{S_t \times d}$ be the text embeddings and $x_{\text{vis}} \in \mathbb{R}^{S_v \times d}$ the projected visual tokens. The transformer input stacks them along the sequence:

$$r^{(0)} = \left[\, x_{\text{text}}; x_{\text{vis}} \,\right]. $$

Unrolling the pre-norm recurrence over $l=1{:}L$ yields the representation read by the LM head,

$$r^{(L)} = r^{(0)} + \sum_{l=1}^{L} a^{(l)} + \sum_{l=1}^{L} f^{(l)}. \quad (4)$$

Eq. 4 shows the additive influences of attention and MLP outputs, but it does not imply disentangled influences: each $a^{(l)}$ and $f^{(l)}$ is a function of the shared stream $r^{(l-1)}$, so changing self-attention alters the inputs that later MLPs receive (and vice versa). Combined with Eqs. 2–3, we can change what concepts are activated by modifying the self-attention (since it feeds into the MLP) or the MLP up/gate layers, and we can change what to write given the activated concepts by changing the down layers. $W_O$ determines the output of the attention layers, but the change in final model output is most influenced by the outputs of the final MLP layers, whichever parameters are tuned (Appendix Fig. 9).

## 3.2 WHICH PARTS TO TUNE?

At the system level, one can update the **vision encoder** and the **projector** (which change $r^{(0)}$), or the **language model** (which produces the layerwise increments that accumulate into $r^{(L)}$). Because $z = U^\top r^{(L)}$ and the sublayers are coupled through the residual, we focus on controlled updates inside the LM that probe *routing* versus *writing* without altering the readout: we keep the LM head $U$, token embeddings, and layer-norm parameters fixed by default.

Guided by Eqs. 2–3, we consider:

- **SA Proj.**: Update $W_Q, W_K, W_V, W_O$ in all blocks (routing + write-back for attention).

- **SA Proj. (QKV)**: Freeze $W_O$ to emphasize routing without directly modifying write-back.

- **MLP**: Update $W_{\text{gate}}, W_{\text{up}}, W_{\text{down}}$ (concept activation + write-back).

- **MLP (Gate & Up)**: Update $W_{\text{gate}}, W_{\text{up}}$ while freezing $W_{\text{down}}$ to regulate write-back.

## 3.3 TRAINING OBJECTIVE

**Target task loss.** We use next-token cross-entropy on the current target dataset with teacher forcing. For a batch $\mathcal{B} \subset \mathcal{D}_{\text{T}}^{(k)}$ at stage $k$,

$$\mathcal{L}_{\text{task}}(\theta) = \mathbb{E}_{(I,y) \sim \mathcal{B}}\left[-\sum_{t=1}^{|y|} \log p_\theta\left(y_t \mid y_{<t}, x_{\text{vis}}\right)\right], \qquad x_{\text{vis}} = g_\psi(f_{\text{vis}}(I)). \quad (5)$$

**Learning-without-Forgetting (optional).** To explicitly curb the output-distribution drift, we can enforce a KL-divergence constraint between the outputs of the current model at stage $k$ with a frozen teacher model (checkpoint after stage $k-1$). Let $\theta_{k-1}$ be the frozen teacher and $\theta$ the current model tuned on $\mathcal{D}_{\text{T}}^{(k)}$. The objective is

$$\mathcal{L}(\theta) = \mathcal{L}_{\text{task}}(\theta) \; + \; \lambda \, \mathcal{L}_{\text{distill}}(\theta; \theta_{k-1}), \qquad (6)$$

with

$$\mathcal{L}_{\text{distill}}(\theta; \theta_{k-1}) = \mathbb{E}_{(I,y)\sim\tilde{\mathcal{B}}}\left[ \frac{\tau^2}{|\mathcal{S}(y)|} \sum_{j\in\mathcal{S}(y)} \text{KL}\Big(\text{softmax}\big(\tfrac{z_{\theta_{k-1},j}}{\tau}\big) \, \| \, \text{softmax}\big(\tfrac{z_{\theta,j}}{\tau}\big)\Big) \right], \quad (7)$$

where $\tilde{\mathcal{B}} \subset \mathcal{D}_{\text{T}}^{(k)}$ is a target minibatch, $\tau$ is the distillation temperature, and $\mathcal{S}(y)$ is a uniformly random subset of positions with $|\mathcal{S}(y)| = \min(|y|, 1000)$ so we distill over many tokens while capping compute/memory. The coefficient $\lambda$ balances fitting the new supervision against preserving the model's earlier behavior.

## 4 EXPERIMENT DESIGN

Our experiments are designed to answer four questions: (i) **Where to tune?**—which components of an LMM can be updated to learn new skills while preserving prior abilities (Sec. 5.1); **Why does forgetting occur?**—whether performance loss is tied to a shift in the model's output distribution (Sec. 5.2); (iii) **How generalizable is our selective tuning strategy**—whether the same selective-tuning recipes (SA Proj., MLP Gate&Up) transfer across model families (Sec. 5.3); and (iv) **How does our selective tuning compare to other simple forgetting mitigation approaches?** (Sec. 5.4).

Due to space limits, we provide details of the tasks and implementation in Appendix A and B.

### 4.1 SEQUENCE-LEVEL METRICS

We summarize performance over the five-stage curriculum with four metrics computed for every method.

- **Target Learning.** At each stage, consider only the task being tuned and measure its improvement over the base model on that task. We then average these stage-wise gains across all stages. This captures how well a method learns the task it is currently trained on.

- **Target Forgetting.** To measure forgetting on target tasks trained earlier in the sequence, we report the average difference between their accuracy immediately after they were trained and their accuracy at the end of the sequence. More negative means more forgetting.

- **Target Overall.** After training the full sequence, we compute the average performance change vs. the base model across all target tasks. This yields the net end-of-sequence effect on the target suite, combining the learned task and the previously learned targets.

- **Held-out Forgetting.** After training the full sequence of target tasks, we measure the average performance across all eight held-out benchmarks, in comparison to the base model. Negative values indicate forgetting on general vision–language ability; positive values indicate positive transfer.

## 5 RESULTS

### 5.1 COMPONENT TUNING ON LLAVA-ONEVISION

Fig. 1 previews learning (single–task tuning, left) and forgetting (held-out along the default sequence, right). Tab. 1 summarizes the four sequence-level metrics on LLaVA-OneVision for each component-tuning configuration, averaged over three five-task curricula. Entries are percentage-point deltas from the base model; the baseline row reports absolute scores. For each column, we run paired sample t-tests on the per-sequence/per-task averages to test whether tuning different components

Table 1: **Effect of tuning different components: learning, forgetting, and overall performance, averaged over three five-task sequences.** Cells are colored using a blue-orange colormap to show performance changes. Blue indicates a positive change, where a darker shade is better. Orange indicates a negative change, where a lighter shade is better. We underline numbers that do not reflect a significantly different task-average distribution from the best, based on a two-sided paired sample t-test.

| Method | Target Learning | Target Forgetting | Target Overall | Held-out Forgetting |
|---|---|---|---|---|
| Baseline | 43.9 | 0.0 | 43.9 | 76.4 |
| Full | +29.9 | −25.9 | +9.2 | −27.4 |
| | − Vision Encoder | +9.6 | −12.7 | −0.5 | −10.8 |
| | − Projector | +2.3 | −0.8 | +1.7 | −1.3 |
| | − LLM | +31.8 | −23.5 | +13.0 | −23.3 |
| | − SA Proj. | +24.9 | −2.3 | +23.1 | −0.6 |
| | − SA Proj. (QKV) | +14.9 | −0.5 | +14.5 | +0.2 |
| | − MLP | +31.1 | −19.5 | +15.5 | −15.7 |
| | − MLP (Gate&Up) | +30.5 | −4.2 | +27.1 | −2.1 |

leads to significantly different per-task learning, forgetting, or overall performance. We underline numbers that are *not* significantly different ($p > 0.1$) than the best. Detailed single-task and sequential results, together with per-task performance tables, are provided in the Appendix.

From the table, we find the following patterns:

**1) Full-model tuning attains large learning but maximizes forgetting**: Target Learning $+29.9$ is coupled with the worst Target/Held-out Forgetting ($-25.9/-27.4$).

**2) Vision-side updates are weak or near-neutral**: the vision encoder yields a modest Target Learning $+9.6$ with negative Target Overall $-0.5$ and Held-out Forgetting $-10.8$; projector-only updates barely move any metric (smallest changes overall).

**3) Language-model tuning has the best learning**: LLM shows the strongest Target Learning $+31.8$ and a solid Target Overall $+13.0$, but also substantial Target/Held-out Forgetting ($-23.5/-23.3$), even better than the Full model.

**4) Self-attention projection is the most stable among LLM choices**: SA Proj. achieves high Target Overall $+23.1$ with minimal forgetting (Target $-2.3$, Held-out $-0.6$); the conservative variant without $W_O$ further reduces forgetting (Target $-0.5$, Held-out $+0.2$) at the cost of learning ($+14.5$ Target Overall), indicating over-regularization when the attention write-back is frozen.

**5) Regulating the MLP write-back offers the best balance**: MLP (Gate&Up) delivers near-maximal Target Learning $+30.5$ and the highest Target Overall $+27.1$ while keeping forgetting small (Target $-4.2$, Held-out $-2.1$); by contrast, full-MLP pushes learning slightly higher on the current task $+31.1$ but increases forgetting (Target $-19.5$, Held-out $-15.7$).

Overall, methods that mainly reroute evidence such as SA Proj. and SA Proj. (QKV), or constrain how activated concepts are written back, such as MLP Gate&Up, provide the most favorable learning–stability trade-off on LLaVA-OneVision.

## 5.2 OUTPUT-DISTRIBUTION PROBE (COUNTING BIAS)

To test whether forgetting is tied to a global shift in token preferences, we track a simple *number-token bias* (NTB) during counting adaptation: as training proceeds, how does the likelihood of outputting a numeric token change for a task that does not require counting? Let $C$ be a fixed subset of vocabulary items (digits and common spelled numerals). For a fixed held-out batch, at training step $s$ we generate a caption with deterministic greedy decoding and, at each step $j$, read the next-token distribution and take the maximum probability over $C$. Averaging first over all positions and then over the batch $\mathcal{B}$ yields

$$\text{NTB}_s = \mathbb{E}_{(I,y)\in\mathcal{B}} \left[ \frac{1}{|y|} \sum_{j=1}^{|y|} \max_{v\in C} p_{\theta_s}(v \mid y_{<j}, x_{\text{vis}}) \right].$$

We plot absolute $\text{NTB}_s$ on a log-spaced grid of steps ($1, 10, 100, 1000$) for SA Proj., MLP, MLP (Gate&Up), MLP (LwF), and full LLM. As seen in Fig. 3, full LLM/MLP sharply increase $\text{NTB}_s$ and

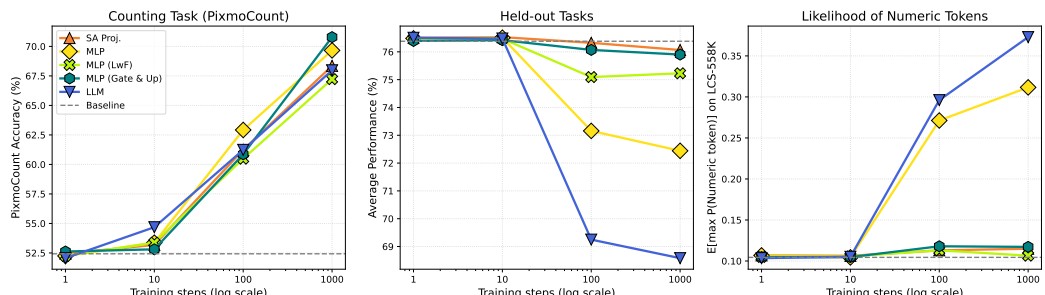

Figure 3: **Learning–forgetting tracks output–distribution shift.** On LLaVA-OneVision tuned for counting, we plot five curves over log-spaced steps for LLM, SA Proj., MLP, MLP (Gate&Up) and MLP (LwF). The dashed line represents the base model. **Left:** PixmoCount accuracy rises for all methods. **Middle:** mean held-out performance drops sharply for LLM and MLP, remains nearly unchanged for SA Proj., and is preserved by MLP (LwF); **Right:** the expected likelihood of number tokens on non-counting captions (LCS-558K (Liu et al., 2023a)) surges for LLM and MLP, stays near baseline for SA Proj., and has little changes for MLP (LwF).

Table 2: Component-level tuning experiments with **LLaVA-NeXT** and **Qwen2.5-VL**. "T" represents "Target" and "H" is for "Held-out". Underlined text of each column denotes the best method.

| Method | LLaVA-NeXT (LLaMA-3 8B) | | | | Qwen2.5-VL (7B) | | | |
|---|---|---|---|---|---|---|---|---|
| | T. Learn | T. Forget | T. Overall | H. Forget | T. Learn | T. Forget | T. Overall | H. Forget |
| Baseline | 31.5 | 0.0 | 31.5 | 59.9 | 52.1 | 0.0 | 52.1 | 77.9 |
| Full | +31.7 | −20.3 | +15.4 | −32.0 | +17.3 | −5.2 | +13.1 | −17.5 |
| | − Vision + Projector | +0.1 | −1.8 | −1.3 | −13.4 | +12.1 | −9.1 | +4.9 | −6.2 |
| | − LLM | +36.2 | −21.2 | +19.3 | −35.9 | +16.8 | −5.9 | +12.1 | −24.6 |
| | − SA Proj. | +28.3 | −7.9 | +21.9 | −7.7 | +16.1 | −1.6 | +14.9 | +0.6 |
| | − MLP | +34.9 | −10.3 | +26.6 | −16.3 | +17.7 | −4.8 | +13.9 | −10.9 |
| | − MLP ($W_{\text{gate}}, W_{\text{up}}$) | +28.0 | −8.9 | +20.9 | −8.7 | +16.8 | +0.4 | +17.1 | −4.6 |

their held-out accuracy drops in tandem; SA Proj. keeps $\text{NTB}_s$ near the baseline with an essentially flat held-out curve; constraining the MLP write-back (tuning only Gate&Up) or distilling to the baseline checkpoint (LwF) suppresses the rise in $\text{NTB}_s$ and correspondingly preserves held-out performance. In our setting, forgetting rises and falls with the magnitude of this shift: updates that mainly *reroute* evidence (SA Proj.) or *regulate write-back* (Gate&Up, LwF) learn the new skill while keeping drift small and thus interference small. We also create an analysis of per-layer contribution of SA Proj. and MLP to the output distribution shift in the Appendix ( Fig. 9), which shows MLP drives the major shift, regardless of what gets tuned.

## 5.3 BEYOND LLAVA-ONEVISION: GENERALIZATION TO OTHER BACKBONES

We repeat the default five-task curriculum on two additional backbones **LLaVA-NeXT (LLaMA-3 8B)** and **Qwen2.5-VL (7B)**, using the same training protocol and sequence-level metrics as for LLaVA-OneVision. For vision-side updates, we tune the vision encoder and projector jointly, since they form a single interface that produces the visual token sequence consumed by the language model.

Across both backbones, the broad picture echoes LLaVA-OneVision: updating the language model is consistently effective for learning new skills; full-model and full-LLM tuning achieve large target task gains but come with the largest drops on held-out. Within the LM, two settings stand out as robust: self-attention projections deliver meaningful target learning with small held-out change, and MLP (Gate&Up) preserves most of the learning of full-MLP while limiting forgetting. There are, however, model-specific nuances worth noting. On **LLaVA-NeXT**, MLP achieves the strongest target-overall improvement but incurs a noticeably larger held-out decrease than SA Proj. or Gate&Up, which remain the most stable choices. LLaVA-NeXT is much more susceptible to forgetting in general than the other models. On **Qwen2.5-VL**, SA Proj. is particularly stable: held-out performance is maintained or slightly improved; MLP (Gate&Up) attains the best target-overall score with near-zero target forgetting; vision + projector tuning also yields non-trivial target gains with moderate stability cost, in contrast to its weaker effect on LLaVA-NeXT. Though Qwen2.5-VL appears to learn less, it

is worth noting that its baseline performance is much higher than that of other models. Overall, taking LLaVA-OneVision, LLaVA-NeXT, and Qwen2.5-VL together, the clearest cross-model takeaway is to prefer SA Proj. when stability on held-out is paramount and MLP (Gate&Up) when seeking near-maximal target learning with limited forgetting; projector-only updates are generally weak, and full-model / full-MLP tuning maximizes short-term gains at a clear stability cost.

## 5.4 MITIGATING FORGETTING

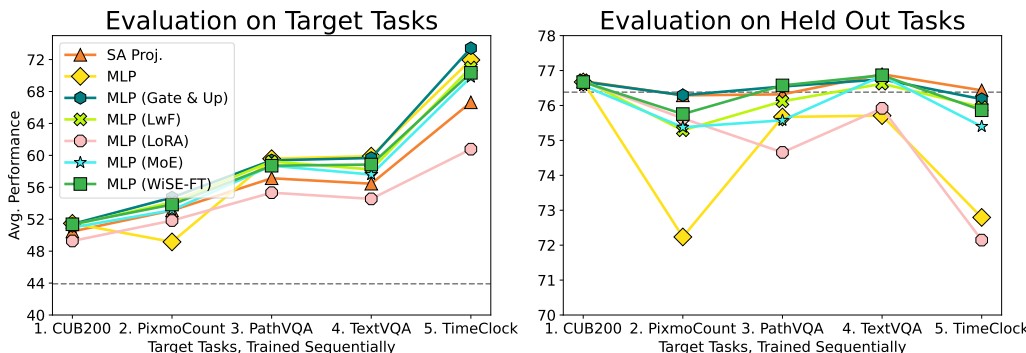

Figure 4: Comparison of different continual learning techniques in the default sequential task curriculum. For LwF, WiSE-FT, only the MLP layers are tuned. LoRA adapters are wrapped only on the MLP layers. MoE is also applied to the MLP layers.

Figure 4 compares three selective tuning recipes, i.e. MLP, SA Proj. and MLP (Gate&Up), with common forgetting mitigation methods: Learning without Forgetting (LwF) (Li & Hoiem, 2017), LoRA, Mixture-of-Experts (MoE), and weight-space ensembling (WiSE-FT) (Wortsman et al., 2022). Refer to the Appendix (Sec. G) for details of these methods. Two patterns emerge. First, SA Proj. and MLP (Gate&Up) provide the best learning–stability trade-off: SA Proj. keeps held-out performance essentially flat while achieving meaningful target gains, and MLP (Gate&Up) delivers stronger target improvements with only a small held-out change, being substantially more stable than MLP. Second, among the compared methods, WiSE-FT can preserve held-out accuracy better than LwF but requires careful selection of task-dependent blending coefficients; LwF reliably curbs forgetting yet may impact target task gains; MoE and LoRA do not match the learning–stability balance of SA Proj. or MLP (Gate&Up), with LoRA often lagging behind on target performance. Overall, selectively tuning SA Proj. or the MLP Gate&Up pair matches or exceeds these mitigation methods while remaining simple (no extra modules, no replay, no per-stage weight blending).

## 6 CONCLUSION

We sought to answer how to teach large multimodal models new skills without erasing prior abilities, and studied this across five target skills, eight held-out benchmarks, and three backbones. Our results show that the apparent loss on held-out tasks after narrow fine-tuning is often *temporary*: performance that drops at one stage can recover later. We trace this behavior to a measurable shift in the next-token distribution rather than the loss of concepts. A simple counting-bias probe makes this drift visible, and a layer-wise residual-to-logit analysis shows that most of the shift is written by late MLP blocks, not by self-attention. Guided by this, we find that tuning only the self-attention projection layers or only the MLP Gate&Up layers limits the bias, leading to good learning with limited forgetting across model families.

Thus, our study helps to understand the learning and forgetting behavior of LMMs, and our recommendations, to limit which components are tuned, are broadly applicable. We hope this work leads to more stable and efficient continuous improvement of LMMs, reducing the environmental and financial cost of model adaptation.

**Limitations.** Due to limited resources, we must leave exploration of many interesting aspects to future work, such as alternative architectures and longer sequences. Also, testing with much larger models, and additional modalities, such as audio, requires further study. Broader issues, such as privacy leakage, safety, and societal impact, remain open for future investigation.

# 7 REPRODUCIBILITY STATEMENT

We aim to make our results fully reproducible. The paper specifies the model backbones and checkpoints (LLaVA-OneVision (Qwen2 7B), LLaVA-NeXT (LLaMA-3 8B), Qwen2.5-VL 7B), the five target tasks and eight held-out benchmarks (Sec. A.1), the sequential curricula, and the training/evaluation settings (Sec. B). Our sequence-level metrics are defined in Sec. 4.1 and used consistently across methods. We provide implementation details, per-task prompts, decoding settings, and numeric token lists in the appendix; method details for the output-distribution probe (Sec. 5.2) are included as supplementary materials. We provide some additional interesting analysis in the appendix as auxiliary information for our discovery path including the layer-wise residual–to–logit analysis, combination of tunable parts, etc. We also provide the detailed per-task performance for both single-task fine-tuning and sequential fine-tuning of each method. We will release code and instructions to fetch datasets and model weights to facilitate end-to-end replication.

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

# Appendix

CONTENTS

# A  TASK DESIGN

## A.1  TASKS AND EVALUATION SUITES

**Target tasks.**  Our criteria for target task selection are: (1) prefer tasks that occur in daily experience, like counting and reading clocks preferred; (2) prefer tasks that LMMs are known to be typically weak, such as fine-grained species classification (Liu et al., 2024a); and (3) exclude tasks used to train the original LLaVA-OneVision model. Based on this, we select **5 target tasks** and create a **default sequential-tuning task curriculum**:

1. **Bird classification** from the CUB dataset (Wah et al., 2011) which contains 5,994 training samples. We reformat the dataset following the instructions of (Liu et al., 2024a) for training and evaluating LMMs.

2. **Counting** from the PixmoCount dataset (Deitke et al., 2025) which contains 36,140 training samples with object count labels.

3. **Medical VQA** from the PathVQA dataset (He et al., 2020) which contains 19,654 radiology question answer pairs.

4. **OCR reading** from the TextVQA dataset (Singh et al., 2019) which has 34,602 training samples.

5. **Time reading** from the TimeClock dataset (Gpiosenka) containing 11,520 training images of analogue clocks with ground truth times.

In total, the curriculum contains 107,910 training samples, providing a comprehensive stress test for forgetting and knowledge transfer.

**Held-out suite.**  To measure generalization beyond the training stream, we evaluate on eight held-out benchmarks: AI2D (Kembhavi et al., 2016), ChartQA (Masry et al., 2022), DocVQA (Mathew et al., 2021), InfoVQA (Mathew et al., 2022), RealWorldQA (visheratin, 2024), SeedBench (Li et al., 2023), ScienceQA (Lu et al., 2022), and MMStar (Chen et al., 2024b). InfoVQA and DocVQA use ANLS; since ANLS is in $[0, 1]$, we average it with accuracies from the other held-out tasks when reporting the mean held-out score.

## A.2  ON BUILDING TARGET TASKS

Our training and evaluation scripts are built on the `LLaVA-NeXT` and `lmms-eval` public GitHub repositories. Since some of the target tasks are not supported by `lmms-eval`, we need to implement support for evaluation of target tasks. Details are as follows.

**Bird classification.**  We reformulate the bird classification dataset CUB200 (Wah et al., 2011) to a multiple choice VQA task following (Liu et al., 2024a). Specifically, for a <image> and <class name> pair, we mix the correct label with 31 randomly chosen labels from the whole dataset and then compose a question like:

```
<image> What species is the bird in this photo?
Answer with the option's letter from the given choices directly.
\n A.<class name A> \n B.<class name> \n ··· Z.<class name Z>
```

This task has 5,794 validation samples. The **instruction prompt** for this task is: *"Answer with the option's letter from the given choices directly."* And only exact match can be deemed as correct by lowercasing model's output and compared to lowercased ground-truth answer.

**Counting.**  The original PixmoCount dataset provides download links rather than actual image files. By the time of downloading, not all links are valid. In the end, besides 36,140 training samples, we collected 535 and 536 validation and test samples. We use the validation set to report numbers in the paper, as done in the technical report of Pixmo dataset (Deitke et al., 2025). The **instruction prompt** for this task is *"Answer with integer and nothing else. For example, if the answer is 1, you should output 1."*. We convert the output by the model to digits and then use exact match to compute accuracy.

Table 3: Parameter groups and counts for LLaVA OneVision Qwen2-7B

| Group | Components | #Params |
|---|---|---|
| $\Theta_{\mathrm{VE}}$ | SigLIP vision encoder | $\approx 400\,\mathrm{M}$ |
| $\Theta_{\mathrm{Proj}}$ | Multimodal projector | $\approx 20\,\mathrm{M}$ |
| $\Theta_{\mathrm{SA}}$ | All blocks: $W_Q, W_K, W_V, W_O$ | $\approx 822\,\mathrm{M}$ |
| $\Theta_{\mathrm{MLP}}$ | All blocks: $W_{\mathrm{gate}}, W_{\mathrm{up}}, W_{\mathrm{down}}$ | $\approx 5{,}703\,\mathrm{M}$ |
| $\Theta_{\mathrm{Emb}}$ | Input token embeddings | $\approx 545\,\mathrm{M}$ |
| $\Theta_{\mathrm{LM}}$ | LM head $U$ | $\approx 545\,\mathrm{M}$ |

[a] SigLIP So400M vision backbone, about 400M parameters.
[b] OneVision Stage 1 projector is about 20M parameters for the 7B class.
[c] Per layer counts with $d$=3584, $d_{kv}$=512, $L$=28: SA 29,364,736, MLP up 135,790,592, MLP 203,685,888; totals multiply by $L$.
[d] Vocab size 152,128 and width $d$=3584 give $152{,}128 \times 3584 = 545{,}226{,}752$ parameters for embeddings and for the LM head (untied).

**Medical VQA.** We use the test split of the PathVQA dataset for evaluation, containing 6,719 samples. The **instruction prompt** for this task is *"For questions that can be answered with a yes or no, just answer yes or no. Otherwise, provide an answer in the medical domain."* We use the exact match score as the metric for this task, using the official evaluation algorithm.

**OCR reading.** `lmms-eval` has support for TextVQA evaluation and we use the accuracy on the validation set as the performance for this task.

**Time reading.** We evaluate on the validation split of the TimeClock dataset, which contains 1,440 samples. The instruction prompt is *"Answer with the time in HH:MM format. For example, if it is 3:45, output 3:45."* To compute accuracy, we parse the model's output to extract the hour and minute. A prediction is marked correct only when both values match the ground truth.

### A.3 TASK CURRICULUM

We provide three task sequences for sequential-tuning:

1. CUB200 → PixmoCount → PathVQA → TextVQA → TimeClock
2. PathVQA → CUB200 → TextVQA → TimeClock → PixmoCount
3. TimeClock → TextVQA → PathVQA → PixmoCount → CUB200

Unless otherwise stated, we use the first as the default sequential-tuning sequence.

## B IMPLEMENTATION DETAILS

### B.1 IMPLEMENTATION DETAILS FOR LLaVA-ONEVISION

We adopt the 7B Qwen2 language model checkpoint for experiments on LLaVA-OneVision. Experiments run primarily on 4×NVIDIA H100 GPUs with 1 sample per GPU. We use 8 gradient-accumulation steps (effective batch 32), a learning rate of $5 \times 10^{-6}$ with cosine decay, and a 3% warm-up. Following the practice from (Li et al., 2024a), we use a smaller learning rate at $2 \times 10^{-6}$ when tuning the vision encoder and the projector. Evaluation uses `lmms-eval` (Zhang et al., 2024) with added support for our targets. When we activate LwF, we set $\lambda=1$ and $\tau = 2$. For all the experiments, we perform single epoch training.

### B.2 PARAMETER COUNT FOR LLaVA-ONEVISION

Tab. 3 lists the parameter groups and counts of each part in the LLaVA-OneVision model. It can be seen that the language model takes a large part of the total capacity of the model. Within the language model, MLP is the major consumer of parameters.

Table 4: Numeric token indices and their corresponding tokens.

| Index / Token | | Index / Token | | Index / Token | | Index / Token | |
|---|---|---|---|---|---|---|---|
| 15 | 0 | 16 | 1 | 17 | 2 | 18 | 3 |
| 19 | 4 | 20 | 5 | 21 | 6 | 22 | 7 |
| 23 | 8 | 24 | 9 | 603 | one | 1960 | ten |
| 3966 | One | 5225 | ONE | 11613 | Two | 14154 | zero |
| 17999 | Zero | 19641 | Three | 19789 | two | 26972 | Four |
| 27856 | three | 32687 | Ten | 34024 | four | 37020 | Five |
| 41460 | Six | 50364 | six | 52670 | five | 58313 | million |
| 59085 | Eight | 59528 | Seven | 67532 | eight | 73956 | ZERO |
| 75796 | Twenty | 80185 | seven | 83329 | Nine | 91602 | Thirty |
| 93223 | nine | 93965 | twenty | | | | |

## B.3 IMPLEMENTATION DETAILS FOR QWEN2.5-VL AND LLAVA-NEXT (LLAMA 3)

We adopt the 7B Qwen2.5 checkpoint for experiments on Qwen2.5-VL. For all experiments on Qwen2.5-VL, we use 4 H100 GPUs and set learning rate at 2e-5 for all components in the model. Per-GPU batch size is set to 4, with 4 gradient accumulation steps. Therefore, the effective batch size is 64.

For experiments on LLaVA-NeXT (LLaMA-3), we use the 8B LLaMA-3 checkpoint. We adopt the same learning rate, warm-up ratio, batch size, gradient accumulation steps as tuning LLaVA-OneVision.

For all the relevant experiments, we perform single epoch training.

## B.4 IMPLEMENTATION DETAILS FOR SEC. 5.2: OUTPUT-DISTRIBUTION PROBE (COUNTING BIAS)

**Setup.** Fix a token subset $C$ (digits and common spelled numerals; exact list in the repo) and a held-out batch $\hat{\mathcal{B}} = \{(I, y)\}$ of $|\hat{\mathcal{B}}| = 100$ image–caption pairs sampled once from LCS-558K (reused for all checkpoints and methods). For each checkpoint $s$ and each $(I, y) \in \hat{\mathcal{B}}$, generate a caption $\hat{y}$ with deterministic greedy decoding using identical preprocessing and decoding settings across methods.

**Per-position score.** At generation step $j$, before committing the token, read the model's next-token probabilities and compute the per-position number tendency

$$p_C^{\max}(\theta_s; I, \hat{y}, j) = \max_{v \in C} p_{\theta_s}(v \,|\, \hat{y}_{<j}, x_{\text{vis}}).$$

**Per-example and batch aggregates.** Summarize each example by the sequence average

$$\text{SeqAvg}_C(\theta_s; I) = \frac{1}{|\hat{y}|} \sum_{j=1}^{|\hat{y}|} p_C^{\max}(\theta_s; I, \hat{y}, j),$$

and aggregate over the batch to obtain the *number-token bias* at checkpoint $s$:

$$\text{NTB}_s = \frac{1}{|\hat{\mathcal{B}}|} \sum_{(I,y) \in \hat{\mathcal{B}}} \text{SeqAvg}_C(\theta_s; I).$$

## B.5 NUMERICAL TOKEN LIST USED FOR COUNTING BIAS PROBE

In Tab. 4, we list the total 38 numeric tokens we used for counting bias probe, and their indices in the tokenizer. The numeric tokens include numeric digits and words such as "one", "ONE", etc.

# C  DISCOVERY PROCESS

In science, the ordering of observation, hypotheses, experimental results, and explanation is important — to know whether claims are post-hoc rationalization of results or experiments are a confirmation of hypotheses that were based on prior observations. Therefore, we wish to give a full accounting.

In beginning this research, we first sought to verify the problem of "catastrophic forgetting" in LMMs. While prior works had largely confirmed the forgetting, these works tending to involve a limited range of tasks, so we created a diverse set of target tasks, some of which we expected to be very hard for the LLaVA-OneVision model (e.g. counting and telling time), and others to be easy (e.g. bird identification and TextVQA). After confirming that typical tuning practices, such as tuning the vision component, LLM, or full model, led to substantial forgetting, we thought we would turn to mitigation strategies, such as experience replay, model expansion with mixture-of-experts, knowledge distillation, and weight averaging. We also noticed a surprising result, that the model performance would drop significantly in held out benchmarks after training on the counting task, it would mostly recover on PathVQA, another specialized task that is not well represented in the benchmarks. Meanwhile, while performing the forgetting mitigation experiments, we also tried separately tuning only the self-attention projection (SA Proj) or MLP layers, motivated by the finding that tuning only the LLM was generally better than tuning the full model. This led to another very surprising result – that tuning only self-attention projection layers led to very good learning of the target tasks with no drop in performance in held out tasks, even after training all five target tasks in a sequence. This was surprising because we were not aware of other instances of strong learning without forgetting behavior, in the absence of model expansion, rehearsal, or strong regularization. A third interesting result was that knowledge distillation turned out to be the most effective method for mitigating forgetting that we tried, outperforming e.g. replay of examples from earlier target tasks and a mixture-of-experts scheme for model expansion.

Initially, we sought to stress test these results. Indeed, we found that if we keep training new tasks, such as the large long-tailed task of iNaturalist (Van Horn et al., 2018) classification, we see a little bit of forgetting. Also, in tuning other models, LLaVA-NeXT and Qwen2.5-VL (Table 2), we do not see exactly the same numbers, of course, but the major trends hold. With Qwen2.5-VL, we actually get a little forward transfer on the held-out tasks when tuning SA Proj. With LLaVA-Next, we get a 7.7 point drop in held out tasks, but less than half the forgetting of tuning MLP layers and less than one-quarter as much as tuning the full LLM. We also tried other sequences of target task training and found that the post-counting recovery of forgetting was not a fluke. For example, we see recovery from both PixmoCount and TimeClock when reversing the sequence order. By-and-large, the results held — fine-tuning self-attention is remarkably robust to forgetting, what was "forgotten" can be recovered without rehearsal, and regularizing the outputs with knowledge distillation also is highly effective in mitigating forgetting when tuning the MLP.

We performed many other experiments, but our breakthrough in understanding came from reviewing the literature, particularly in work, such as Geva et al. (Geva et al., 2021) and Olsson et al. (Olsson et al., 2022), that experimentally explore the roles of transformer components. Their key results are that MLPs are responsible for storing and applying memories, with the up layer(s) looking up the memories (or activating concepts) and the down layers applying the activated concepts to modify the output token distribution. Attention, on the other hand, is responsible for processing and organizing the inputs. This led us to consider that a model can adapt to a task in many ways: acquiring skills to make better use of its inputs, acquiring new memories and concepts, better applying those concepts, or simply biasing toward the output distribution. We hypothesized that, when training the full LLM, the model is at least partially taking a shortcut to bias toward the output distribution, rather than focusing on skill or memory improvement. This hypothesis could explain all three observed phenomena. The SA Proj is robust to forgetting because it does not directly tune the MLP layers that produce the output distribution. The forgetting is sometimes recoverable because subsequent training on a task with more varied outputs reverses the narrow output distribution shift. Knowledge distillation directly penalizes shift in the output distribution.

This led us to propose two experiments to test this hypothesis. First, we reasoned, we should see that, as the counting task is trained, the model becomes more predisposed to output numbers, since the counting task answers are always of the form, "There are [number] [object(s)] in this image." We also should see some correlation between this bias toward numeric tokens and forgetting in held-out

benchmarks. As we show in Fig. 3 and exemplify in Fig. 11, the results are quite striking with a strong effect of the output distribution bias and a strong correlation with forgetting. Second, we proposed, tuning the MLP except for the down layers that most directly modify the output distribution should mitigate the output bias and, therefore, reduce forgetting. Again, the confirmation was strong — with LLaVA-OneVision, tuning only MLP up layers achieved the best overall target performance with only a little more forgetting than tuning SA Proj.

We believe our results are conclusive, especially given the observe-hypothesize-test-confirm pattern of our research. Further, in the past few days (at the time of this writing), another paper has come out with related findings. Shenfield et al. (Shenfeld et al., 2025) finds that the amount of forgetting is correlated to distributional shift between the base and tuned model, as measured by KL-divergence, and this explains why on-policy RL training is more robust to forgetting than SFT (supervised fine-tuning).

Finally, we would like to stress that our experiments have been more thorough than we can relate in the main text. Just in generating the results of Table 1, we fine-tuned the 7B parameter model on 5 tasks 21 times (3 sequences, 7 components) and evaluated 8 broad benchmarks and 5 target tasks 105 times (after each target task was trained). That is 105 task trainings and 1365 task evaluations. We include many other experimental results in the main text and appendix below. While it is always possible to train more models, more component variations, more mitigation strategies, on more datasets and with more evaluations, we have pushed our resources to the brink and hope that the reader finds our claims sufficiently supported, as we do.

## D    USE OF LARGE LANGUAGE MODELS (LLMS)

We used a general-purpose large language model (LLM) as an assistive tool for *writing and editing*. Specifically, the LLM helped (i) refine phrasing for the abstract, introduction, method descriptions, and result summaries; (ii) advise on LaTeX syntax and commands, and help address compile errors; and (iii) brainstorm alternative framings and terminology for clarity and coherence. The LLM also proposed suggestions for paper organizations.

The LLM *did not* design or run experiments, collect data, produce numerical results, or generate figures. All experimental protocols, scripts, hyperparameters, and evaluations were implemented by the authors; all numbers and tables in the paper are computed from our own training/evaluation runs. When the LLM proposed text for technical definitions or claims, we verified the statements against our code, logs, and checkpoints and revised as needed. All citations were added and checked by the authors.

No private or sensitive data were shared with the LLM beyond draft text and public references. Final responsibility for the content rests with the authors.

## E    MORE RESULTS

### E.1    SINGLE TASK FINE-TUNING

In Fig. 1, we show the learning and forgetting of tuning different components on one target task at a time, and then recording the performance for that target task and the average held-out performance. In Tab. 5 we show the actual performance of each component by taking the delta based on the baseline (original model), ordered from least to most parameters.

As a general trend, tuning more parameters increases both *learning* (improvement in target task) and *forgetting* (decrease in average held-out tasks), with vision encoder and self-attention projection as the notable exceptions. Tuning the **full** network or only the **language model** yields the greatest learning (+33.3 and +31.8 percentage points, on average), yet these gains are accompanied by significant forgetting (-3.2 and -2.8 points). Adjusting the **MLP** of the LLM provides a good trade-off, with similar learning (+31.6) and substantially lower forgetting (-1.4). Adjusting only the **self-attention projection layers** achieves a respectable +24.2 in learning and, surprisingly, *no measurable forgetting*. The **vision encoder** and the **projector** offer relatively little gain, and the vision encoder has the most forgetting (-4.2), indicating that tuning the vision features is particularly disruptive.

Table 5: **Single task fine-tuning by component.** Each individual target task is fine-tuned from the original model, and the performance for that task ("Target") and average held-out performance ("Held-out") is measured. Each row is for tuning a different component or set of components: "Proj.", "Vis. Enc.", "SA Proj.", "MLP", "LM", "Full" represent tuning the projector, vision encoder, self-attention projections in the LLM, MLP in LLM, full LLM, and all parameters, respectively. "+" is an increment over the baseline (original LLaVA-OneVision-7B checkpoint), and "-" is a decrease.

| Method | CUB200 | | PixmoCount | | PathVQA | | TextVQA | | TimeClock | | Average | |
|---|---|---|---|---|---|---|---|---|---|---|---|---|
| | Target | Held-out | Target | Held-out | Target | Held-out | Target | Held-out | Target | Held-out | Target | Held-out |
| Baseline | 53.7 | 76.4 | 52.4 | 76.4 | 36.3 | 76.4 | 76.0 | 76.4 | 1.1 | 76.4 | 43.9 | 76.4 |
| Proj. | +5.7 | -0.0 | +4.2 | -0.1 | +0.6 | -0.1 | +0.5 | +0.2 | +0.4 | -0.5 | +2.3 | -0.1 |
| Vis. Enc. | +16.1 | -0.8 | +11.6 | -4.7 | +3.7 | -2.8 | +1.0 | -0.7 | +12.7 | -11.9 | +9.0 | -4.2 |
| SA Proj. | +31.8 | +0.3 | +15.2 | -0.2 | +14.4 | -0.3 | +3.5 | +0.3 | +56.0 | -0.1 | +24.2 | +0.0 |
| MLP | +36.4 | +0.3 | +17.8 | -4.0 | +26.5 | -0.4 | +3.8 | +0.0 | +73.3 | -3.1 | +31.6 | -1.4 |
| LM | +40.0 | -0.0 | +16.3 | -7.7 | +26.8 | -0.7 | +3.5 | -0.7 | +72.6 | -4.6 | +31.8 | -2.8 |
| Full | +37.0 | +0.1 | +19.0 | -9.0 | +27.4 | -0.9 | +3.4 | -0.7 | +79.8 | -5.4 | +33.3 | -3.2 |

Now, consider the variations by task. There is only a weak correlation between the amount learned and forgotten per task. For instance, CUB200 has the second-most learning (after TimeClock) but the least forgetting. Also, some tasks benefit from visual tuning while others do not. Fine-grained bird recognition (CUB200) and medical question answering (PathVQA) benefit almost exclusively from language model updates, gaining +40.0 and +26.8 points, respectively, with little or no benefit from additional vision tuning. Conversely, for PixmoCount and TimeClock, tuning the full model handily outperforms tuning only the LLM portion.

### E.2 SEQUENTIAL FINE-TUNING

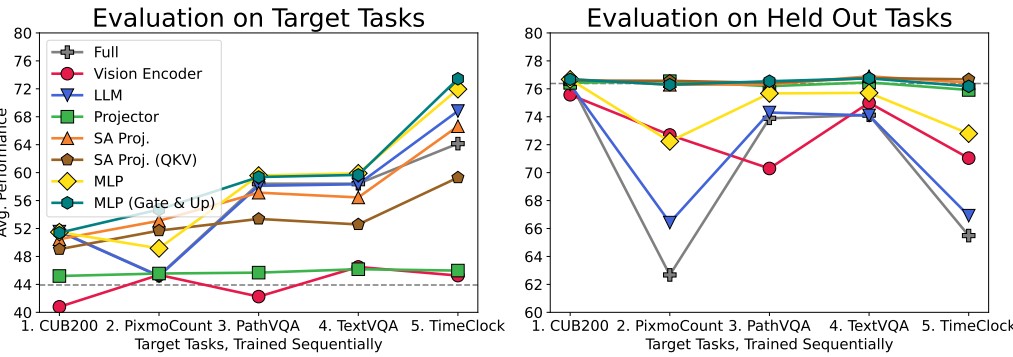

Figure 5: **Sequential fine-tuning by component.** The target tasks in the $x$-axis are trained sequentially, from left to right. After training each task, the average performance of all target tasks (**left**) and all held-out tasks (**right**) are measured. Each line shows the performance after tuning a different component or set of components: LLaVA-OneVision (Full, Vision Encoder, Projector, LLM, SA Proj., SA Proj. (QKV),, MLP, MLP (Gate&Up)). The dashed horizontal gray line marks the average performance of the original model.

In Fig. 5, we display how sequentially tuning different components on the default sequence of all five tasks affects the *average performance of all target tasks* and held-out tasks. In this case, forgetting in later learning can affect the performance of target tasks learned earlier. Per-task performance is attached in the later sections.

Updating the MLP (Gate&Up) gives the best target-task performance overall. Multiple methods have stable results on held-out tasks through out the whole sequence, such as SA Proj., SA Proj. (QKV), and MLP (Gate&Up).

Another interesting phenomenon which is also mentioned in the main paper is that held-out performance does not continually drop as more tasks are trained, but rises and falls. For example, training on PixmoCount causes substantial loss in held out performance (0.76 to 0.63 for the full model), but

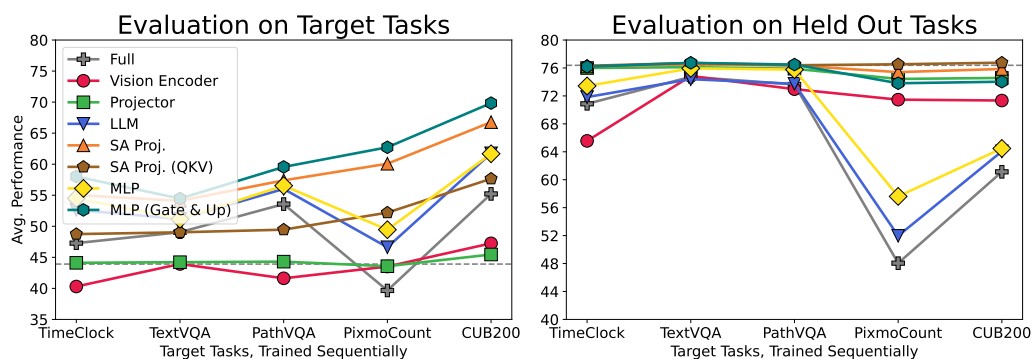

Figure 6: **Sequential fine-tuning by component.** Tasks are arranged as TimeClock → TextVQA → PathVQA → PixmoCount → CUB200.

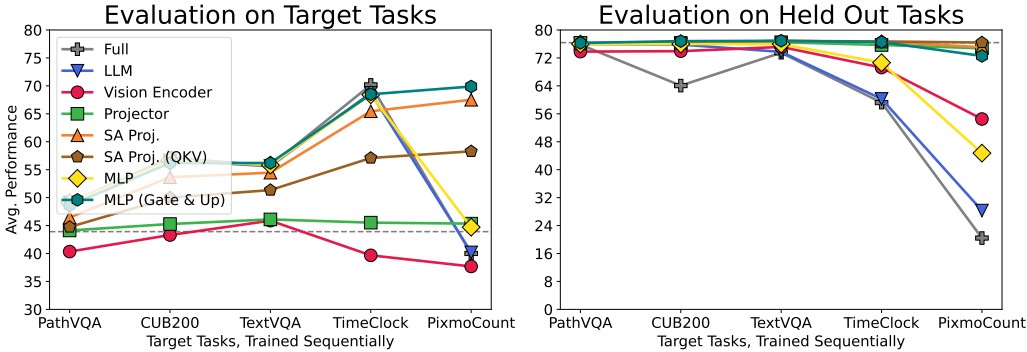

Figure 7: **Sequential fine-tuning by component.** Tasks are arranged as PathVQA → CUB200 → TextVQA → TimeClock → PixmoCount.

the loss is largely recovered (to 0.74) by training the next task PathVQA. This means in "forgetting", much of the information is not permanently lost but temporarily inaccessible.

In Figs. 7 and 6, we show the sequential tuning results by component on LLaVA-OneVision in the other two orders. Fig. 6 validates that forgetting recovery is not order-specific: methods that forget significantly on PixmoCount, rebound after tuning on CUB200. Both figures indicate the robustness of SA Proj., SA Proj. (QKV), and MLP (Gate&Up) on held-out tasks as they essentially keep flat throughout. Especially, MLP (Gate&Up) has a huge benefit in target learning.

# F    More analysis

## F.1    Composing stable tuning strategies: SA Proj. + MLP Gate&Up

We asked whether combining the two most stable, high-learning settings from the main paper, i.e., SA Proj. and MLP (Gate&Up), has further benefits. We evaluate two compositions: **SA Proj. + MLP (Gate&Up)** and **SA Proj. (QKV only) + MLP (Gate&Up)** under the same five-task curriculum, reporting the same sequence-level metrics and counting-bias probe. In aggregate, the composed variants match or slightly underperform the two standalone settings on target learning while keeping held-out changes small; the QKV-only composition is better than the other composition in the held-out performance. But across tasks and checkpoints, neither composition consistently dominates MLP (Gate&Up) alone, indicating that most of the achievable learning–stability trade-off is already realized by Gate&Up for LLaVA-OneVision.

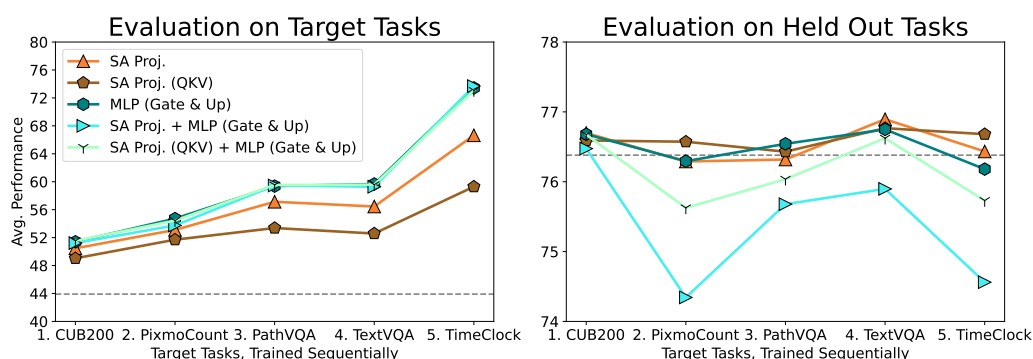

Figure 8: **Composing stable updates.** We compare SA Proj. and MLP (Gate&Up) to two compositions: **SA Proj. + Gate&Up** and **SA Proj. (QKV only) + Gate&Up** using the default five-task sequential tuning.

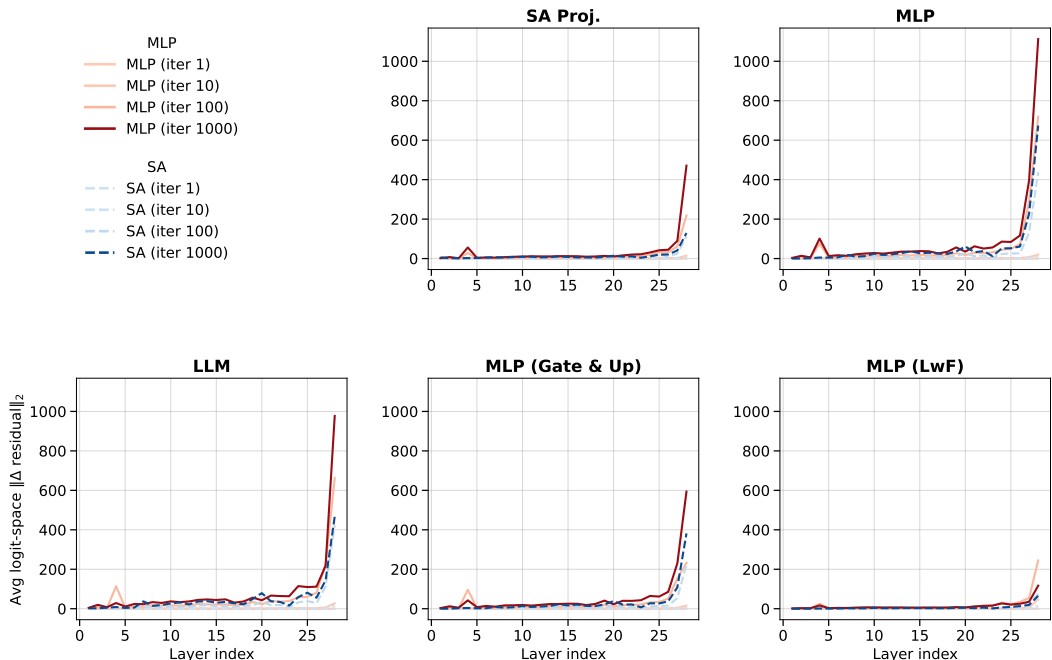

Figure 9: Layer-wise residual-delta magnitude across training iterations. Each subplot shows the average logit-space $\|\Delta\text{residual}\|_2$ attributable to MLP (solid reds) and self-attention projections (dashed blues) versus transformer layer index. The five method configurations are: SA Proj., MLP, LLM, MLP (Gate & Up), and MLP (LwF). Color shade encodes checkpoint iteration (darker = later). The top-left panel is a legend; other panels omit legends for clarity.

### F.2 LAYER-WISE RESIDUAL−TO−LOGIT CONTRIBUTION ANALYSIS

We quantify where (by depth) and how strongly (by pathway) adaptation perturbs the output distribution by comparing the *logit-space* effects of self-attention versus MLP residual updates across layers and training steps. We evaluate on a fixed, held-out multimodal shard sampled once from LCS-558K and reuse it for all methods and checkpoints; unless noted, statistics are computed under teacher forcing over the assistant answer span (target tokens). For each tuning configuration (SA Proj., MLP, LLM, MLP (Gate&Up), and MLP (LwF)) we compare tuned checkpoints to the frozen stage-0 base model at log-spaced training steps (e.g., $1, 10, 100, 1000$), excluding the combined SA Proj. + MLP (Gate&Up) condition. We register forward hooks on every decoder layer's self-attention and MLP submodules in both the base and tuned models to capture their residual increments $a^{(l)}$ and $f^{(l)}$ at each token $j$; with the LM head $U$ fixed, we form logit-space deltas by projecting the *difference* of

residual contributions through $U$:

$$\Delta z_{\text{SA}}^{(l)}(j) = U^\top \big(a_{\text{tuned}}^{(l)}(j) - a_{\text{base}}^{(l)}(j)\big), \qquad \Delta z_{\text{MLP}}^{(l)}(j) = U^\top \big(f_{\text{tuned}}^{(l)}(j) - f_{\text{base}}^{(l)}(j)\big).$$

For each layer we aggregate token-wise vectors into a scalar via the $\ell_2$ norm and then average across tokens and examples to obtain per-layer logit-space magnitudes:

$$\text{SA}(l) = \sqrt{\mathbb{E}_j\big[\|\Delta z_{\text{SA}}^{(l)}(j)\|_2^2\big]}, \qquad \text{MLP}(l) = \sqrt{\mathbb{E}_j\big[\|\Delta z_{\text{MLP}}^{(l)}(j)\|_2^2\big]}.$$

We report these two curves (dashed blue for self-attention, solid red for MLP) per checkpoint, sharing axes across panels for comparability.

**Key observations.** (1) **MLP dominates the shift.** Across configurations and steps, MLP curves exceed self-attention curves—often by $>2\times$ in later layers—indicating that most logit-space change comes from the MLP pathway. (2) **Drift grows with training.** For settings that forget (e.g., full LLM or full MLP), per-layer magnitudes increase monotonically with checkpoint step, mirroring the counting-bias rise and held-out decline. (3) **Late layers drive the effect.** The last 4–5 transformer layers account for the vast majority of the shift, with the final two layers contributing the largest deltas; early layers remain comparatively stable. (4) **Regulating write-back reduces drift.** MLP (Gate&Up) and MLP (LwF) substantially shrink late-layer MLP magnitudes relative to full MLP, aligning with their smaller held-out drops. (5) **Self-attention changes are smaller and flatter.** SA Proj. curves are consistently below the corresponding MLP curves and vary less across steps, indicating weaker and less step-sensitive contribution to the overall distribution shift.

### F.3 Effect of tuning different LLM layers

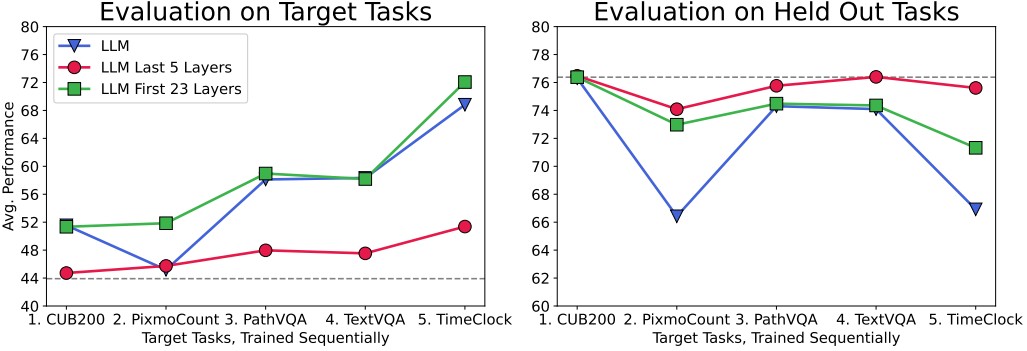

Figure 10: **Sequential tuning of LLM layer subsets.** We compare tuning the full LLM against tuning only the first 23 layers or only the last 5 layers of the 28-layer Qwen2-7B model.

Guided by the observation that late transformer layers are the primary drivers of output distribution shift (Sec. F.2), we investigate how different layer subsets contribute to learning versus forgetting. We partition the 28-layer Qwen2-7B language model into two subsets: the **First 23 Layers** (blocks 0–22) and the **Last 5 Layers** (blocks 23–27). We perform sequential fine-tuning on our default five-task curriculum, updating only one subset while freezing the rest.

We can read the following from Fig. 10:

- **Tuning only the *Last 5 Layers* yields minimal learning gains.** This suggests the late layers alone lack the capacity to adapt to new tasks, and as a consequence, they also induce little forgetting.
- **Tuning only the *First 23 Layers* drives strong learning,** achieving target task performance comparable to, and at times exceeding, tuning the full LLM. Because the *Last 5 Layers* are frozen in this "First 23" experiment, the forgetting seen in the full LLM is largely alleviated. The sharp drop from the counting task (PixmoCount) is significantly mitigated.

In summary, these results directly confirms our analysis that constraining the output distribution drift is an effective strategy to mitigate forgetting.

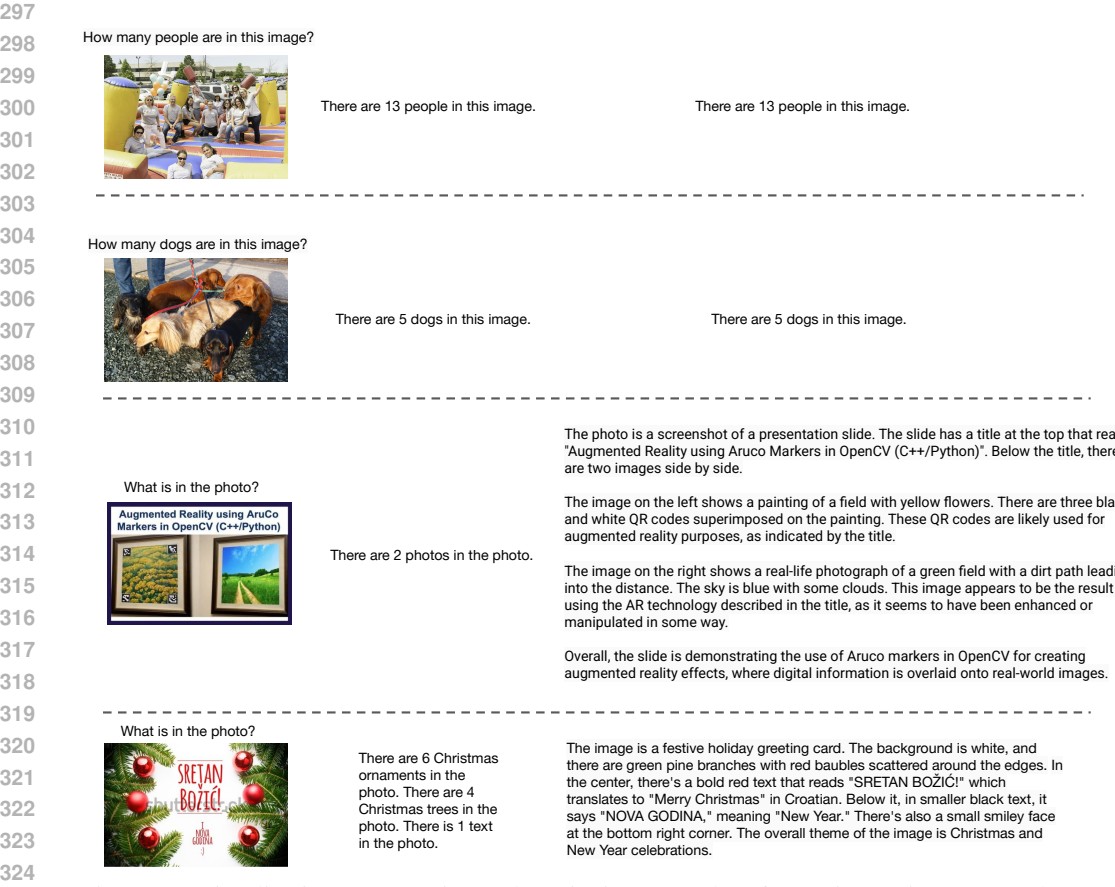

Figure 11: Visualizations on counting and captioning examples after tuning tuning MLP and SA Proj. on the counting task. The counting and examples are sampled from the PixmoCount dataset and the LCS-558K (Liu et al., 2023a) dataset, respectively.

### F.4 QUALITATIVE RESULTS OF MLP AND SA PROJ. TUNING

In Fig. 11, we demonstrate the response differences between tuning the MLP and tuning SA Proj. in the LLM on PixmoCount for 1K iterations. Hence, counting examples can be regarded as the target evaluation and we draw two image captioning samples from LCS-558K (Liu et al., 2023a) as a held-out evaluation. It can be seen that SA Proj. can both output the correct answers for counting examples and remain the capability to give detailed responses when asked to caption images. As a contrast, MLP uses the learned counting skill to describe the image contents, for example, "There are 2 photos in the photo." for the third row. It demonstrates that MLP temporarily forgets how to answer this question but still remains the capability to conceptualize image contents and recognize objects.

### F.5 TRAINING EFFICIENCY COMPARISON OF DIFFERENT FORGETTING MITIGATION METHODS

We compare model size, number of trainable parameters, and training speed for the forgetting mitigation methods shown in Fig. 6. The results appear in Tab. 6. All experiments were run on four NVIDIA H100 96 GB GPUs with DeepSpeed in `bfloat16`, and no competing processes were active. Each run processed exactly 384 training samples. **Total Params** and **Trainable Params** are reported in billions. **Train SPS** reports the average number of training samples processed per second, reflecting data loading and optimization steps.

We observe the following:

- **SA Proj.** achieves the highest throughput at 1.46 samples / sec. while using only 0.82 B trainable parameters.

Table 6: Training efficiency and parameter footprint of the evaluated model variants ($4 \times$ H100 96 GB, DeepSpeed, bfloat16). **Total Params** is the total number of model parameters in billions. **Trainable Params** is the subset that requires gradient updates in billions. **Train SPS** is the number of training samples processed per second, collected from training models on the same 384 samples.

| Method | Total Params (B) | Trainable Params (B) | Train SPS ↑ (#samples / sec.) |
|---|---|---|---|
| SA Proj. | 8.03 | 0.82 | 1.46 |
| MLP (Gate&Up) | 8.03 | 3.80 | 1.45 |
| MLP | 8.03 | 5.70 | 1.44 |
| LoRA | 8.51 | 0.50 | 1.27 |
| LwF | 16.06 | 5.70 | 0.81 |
| MoE | 13.73 | 5.70 | 0.44 |

- **MLP (Gate&Up)** follows closely at 1.45 samples / sec. but requires larger number of trainable parameters (3.8 B).

- **MLP** uses more trainable parameters (5.7 B) though the train SPS is very close to the above two variants.

- **LoRA** uses the fewest trainable parameters at 0.5 B, though its throughput is lower at 1.27 samples / sec. due to processing every token through adapter weights.

- **LwF** maintains a teacher network, resulting in the largest total parameter count of 16.06 B and a reduced throughput of 0.81 samples / sec. because of the extra forward pass and distillation loss.

- **MoE** introduces two experts per MLP module and a layerwise gating network, inflating total parameters to 13.73 B and reducing throughput further to 0.44 samples / sec.

## F.6 OUTPUT DISTRIBUTION DYNAMICS ACROSS FORGETTING AND RECOVERY

To understand the output distribution dynamics during both forgetting and subsequent recovery, we conducted an additional analysis. Building upon the methodology of our counting-bias probe (Sec. 5.2 and Sec. B.4), we extended this analysis across three sequential stages of our curriculum: initial training on CUB200, followed by PixmoCount (where forgetting typically occurs), and finally PathVQA (where recovery is observed).

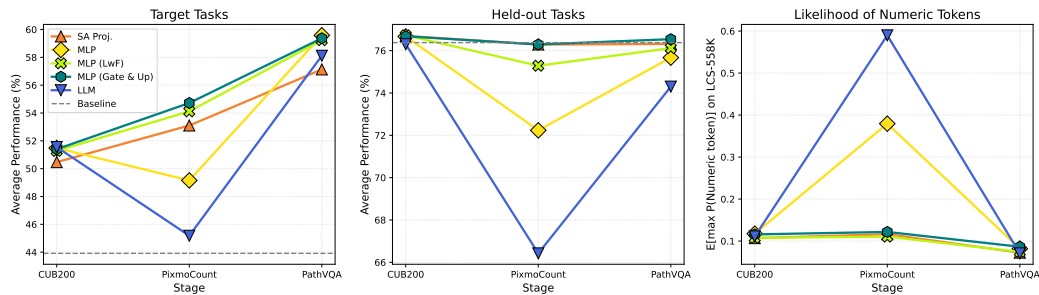

Figure 12: **Analysis of the forgetting and recovery cycle over three sequential tasks.** We plot performance on **(Left)** average target tasks, **(Middle)** average held-out tasks, and **(Right)** the likelihood of numeric tokens on a general held-out dataset (LCS-558K). The plots track performance after training on CUB200 (Stage 1), PixmoCount (Stage 2), and PathVQA (Stage 3). The key finding is in the rightmost plot: for methods susceptible to forgetting (e.g., LLM, MLP), the numeric token bias (output shift) **spikes** at Stage 2 (correlating with the performance drop) and then **recedes** at Stage 3 (correlating with the performance recovery). This provides direct quantitative evidence that the "recovery" phenomenon is a re-correction of the output distribution shift.

The results are presented in Fig. 12. The rightmost plot directly visualizes the output distribution shift for numeric tokens and its recovery for different fine-tuning methods. Comparing this with the middle plot for the held-out performance, we observe:

- **Forgetting (PixmoCount):** For methods susceptible to forgetting (e.g., LLM Full, MLP), training on PixmoCount leads to a significant drop in held-out performance (middle plot) and a simultaneous spike in the likelihood of numeric tokens (right plot). This re-confirms our finding that forgetting is directly correlated with an output distribution shift towards the task-specific bias.
- **Recovery (PathVQA):** Crucially, after training on PathVQA, which requires diverse, non-numeric outputs, we observe a clear recovery in held-out performance for these same methods. This recovery is directly mirrored by a reduction in the likelihood of numeric tokens, bringing the output distribution closer to its pre-counting state.

This extended probe explicitly demonstrates that the "recovery" of held-out performance is indeed a consequence of the model's output distribution shifting away from the previous task's bias and back towards a more general, un-biased state. This provides strong evidence for the mechanism of recoverable output distribution shift throughout the entire forgetting and recovery cycle. We note, however, that the held-out performance does not fully recover to its original baseline. We hypothesize that, this is likely because the PathVQA task, while more general than counting, still introduces its own subtle, task-specific distribution shifts on other tokens. This new bias prevents a perfect return to the original general-purpose state, even as it corrects the more damaging numeric bias from the previous task. Quantifying these more complex, overlapping distribution shifts remains a challenging direction for future work.

## F.7 QUALITATIVE ANALYSIS OF FORGETTING AND RECOVERY

| Input image | Method | Baseline | CUB200 | PixmoCount | PathVQA |
|---|---|---|---|---|---|
| | LLM | A white bed with a tan headboard and white pillows. | A bed with a white mattress and a blue pillow on it. | There is 1 pillows in this image. | A white bed with a tan headboard |
| | MLP | A white bed with a tan headboard and white pillows. | A white bed with a tan headboard and a blue pillow. | There is 1 pillows on the bed. | A white bed with a tan headboard |
| | MLP (Gate&Up) | A white bed with a tan headboard and white pillows. | A white bed with a tan headboard and a blue pillow. | A bed with a white mattress and a blue pillow. | A bed with a white mattress and a blue pillow. |
| | MLP (LwF) | A white bed with a tan headboard and white pillows. | A white bed with a tan headboard and white pillows. | A bed with a white mattress and a blue pillow. | A bed with a white mattress and a blue pillow on it. |
| *Q: Render a clear and concise summary of the photo.* | SA Proj. | A white bed with a tan headboard and white pillows. | A white bed with a tan headboard and white pillows. | A bed with a white mattress and a blue pillow. | A white bed with a tan headboard and a blue pillow. |
| | LLM | A pair of knitted boots with orange soles. | A pair of grey and orange knitted boots. | There are 2 boots in this image. | A pair of grey and orange knitted boots |
| | MLP | A pair of knitted boots with orange soles. | A pair of knitted boots with orange soles. | There are 2 boots that are grey and orange. | Knit boots with white and orange trim |
| | MLP (Gate&Up) | A pair of knitted boots with orange soles. | A pair of knitted boots with orange soles. | A pair of grey and orange knitted boots. | A pair of knitted boots in grey and orange with white trim. |
| | MLP (LwF) | A pair of knitted boots with orange soles. | A pair of knitted boots with orange soles. | A pair of knitted boots with orange soles. | A pair of knitted boots with orange soles |
| *Q: Describe the image concisely* | SA Proj. | A pair of knitted boots with orange soles. | A pair of knitted boots with orange soles. | A pair of grey and orange knitted boots. | A pair of knitted boots in grey and orange with white trim. |

Figure 13: Results of different models on captioning examples from the LCS-558K (Liu et al., 2023a) dataset after sequentially fine-tuning on CUB200, PixmoCount, and PathVQA.

In addition to the quantitative probes, Figure 13 provides a direct qualitative visualization of the forgetting and recovery cycle. This figure displays model responses to general, held-out image captioning prompts (e.g., "Describe the image concisely") at key sequential stages: Baseline, after CUB200, after PixmoCount (the "forgetting" stage), and after PathVQA (the "recovery" stage).

The results clearly illustrate the output shift mechanism. We observe two key behaviors in the methods most prone to forgetting (**LLM** and **MLP**):

- **Forgetting (After PixmoCount):** After training on the narrow counting task, these models exhibit a strong output bias. They incorrectly reframe the general captioning request as a counting problem,

producing outputs like "There are 2 boots in this image". This qualitatively demonstrates the "forgetting" as a task-specific output shift, not a loss of core concepts (the model still identifies "boots").

- **Recovery (After PathVQA):** Crucially, in the final column, after training on the more diverse PathVQA task, these same models **regain their captioning ability**. The output bias is corrected, and they once again produce the correct, descriptive caption (e.g., "A pair of grey and orange knitted boots").

This provides a powerful, intuitive visualization to confirm the hypothesis: "forgetting" is likely a temporary output distribution shift, and "recovery" is the re-correction of that shift by a subsequent task's training signal.

## G  FORGETTING MITIGATION APPROACH DESCRIPTIONS

**Low-rank adaptation (LoRA).**    When memory or compute restricts full fine-tuning, LoRA (Hu et al., 2022) offers a lightweight alternative. For a weight matrix $W_0 \in \mathbb{R}^{d \times k}$, we introduce two trainable low-rank matrices $A \in \mathbb{R}^{r \times k}$ and $B \in \mathbb{R}^{d \times r}$ and model the update as $\Delta W = BA$. After optimization, the effective weight becomes

$$W = W_0 + \tfrac{\alpha}{r} BA,$$

where $\alpha$ is a scalar scaling factor. Because only $A$ and $B$ are updated, the number of learned parameters per task drops from $dk$ to $r(d + k)$, which is substantial when $r \ll \min(d, k)$.

In the continual-learning setting, we instantiate a fresh pair $(A^t, B^t)$ for each task $\mathcal{T}_t$ while keeping the backbone weights frozen. After completing task $\mathcal{T}_t$, we merge the low-rank update into the backbone weight $W_t \leftarrow W_{t-1} + \tfrac{\alpha}{r} B^t A^t$ and then discard the adapters. This maintains a *constant* parameter footprint across tasks and avoids accumulating a growing set of task-specific modules.

**Weight-space interpolation**    Weight-space interpolation (Wortsman et al., 2022) forms an implicit ensemble by linearly combining the pretrained/base checkpoint with the fine-tuned checkpoint. Given the base weights $W_{\mathrm{base}}$ (the original LLaVA-OneVision checkpoint) and the fine-tuned weights after stage $t$, $W_t^{\mathrm{FT}}$, we build an interpolated model

$$W_t^{(\beta)} = (1 - \beta) W_{\mathrm{base}} + \beta W_t^{\mathrm{FT}}, \quad \beta \in [0, 1]. \tag{8}$$

The coefficient $\beta$ trades off specialization on the current target task (larger $\beta$) against retention of general capabilities (smaller $\beta$).

In our sequential setting, we apply Eq. equation 8 *after* finishing fine-tuning on task $\mathcal{T}_t$ and evaluate $W_t^{(\beta)}$. Unless otherwise noted, optimization for the next stage continues from $W_t^{\mathrm{FT}}$ (not from $W_t^{(\beta)}$) to avoid repeatedly biasing training toward the base weights. We sweep $\beta \in \{0.1, 0.3, 0.5, 0.7, 0.9\}$ and report $\beta{=}0.3$'s result for comparison as it leads to better learning and forgetting tradeoff compared to results obtained from other $\beta$-s.

**Mixture of Experts.**    We next leverage the *Mixture of Experts* (MoE) architecture to expand model capacity without overwriting knowledge learned during pretraining (Wei et al., 2024). An MoE layer combines a set of specialist networks (experts) $\{E_i\}_{i=1}^N$ through a learnable *gating network* $g$ that produces input dependent weights. The layer output is

$$l = \sum_{i=1}^{N} g_i(x) E_i(x),$$

typically with a sparsity constraint such as top-$k$ gating so that only a few experts are active per input. We follow standard practice and replace the feed-forward (MLP) submodule in every transformer decoder block of the language model with an MoE layer.

At the start of continual training, each decoder block contains 1) the **pretrained expert** $E_{\mathrm{pt}}$ that stores upstream knowledge and 2) a new **tuned expert** $E_{\mathrm{new}}$ that is a copy of $E_{\mathrm{pt}}$. The gating network is a linear layer initialized to all zeros, which initially routes the entire token sequence

Table 7: Detailed performance of using **MoE** to mitigate forgetting by performing sequential fine-tuning on each target task.

| Dataset | Baseline – | Stage 1 CUB200 | Stage 2 PixmoCount | Stage 3 PathVQA | Stage 4 TextVQA | Stage 5 TimeClock |
|---|---|---|---|---|---|---|
| **Target** | | | | | | |
| CUB200 | 53.7 | 87.7 | 87.6 | 87.4 | 87.3 | 87.0 |
| PixmoCount | 52.4 | 53.9 | 65.5 | 68.2 | 63.5 | 59.9 |
| PathVQA | 36.3 | 36.2 | 35.4 | 61.3 | 57.4 | 57.8 |
| TextVQA | 76.0 | 76.0 | 75.3 | 75.3 | 78.5 | 76.3 |
| TimeClock | 1.1 | 1.0 | 1.9 | 1.2 | 1.2 | 68.0 |
| **Average** | 43.9 | 51.0 | 53.1 | 58.7 | 57.6 | 69.8 |
| **Held out** | | | | | | |
| AI2D | 81.4 | 81.5 | 81.2 | 80.5 | 81.3 | 80.8 |
| ChartQA | 80.1 | 80.2 | 79.6 | 79.6 | 80.0 | 76.1 |
| DocVQA | 87.1 | 87.2 | 85.2 | 85.8 | 86.6 | 84.5 |
| InfoVQA | 65.9 | 65.6 | 64.0 | 64.8 | 66.4 | 64.4 |
| MMStar | 61.8 | 62.1 | 62.6 | 62.6 | 62.5 | 61.4 |
| RealWorldQA | 66.4 | 67.7 | 62.2 | 63.9 | 69.3 | 68.0 |
| ScienceQA | 95.9 | 95.8 | 95.9 | 95.7 | 96.3 | 96.0 |
| SeedBench | 72.4 | 72.5 | 72.2 | 71.6 | 72.4 | 72.1 |
| **Average** | 76.4 | 76.6 | 75.4 | 75.6 | 76.8 | 75.4 |

Table 8: Detailed performance of using **LoRA** to mitigate forgetting by performing sequential fine-tuning on each target task.

| Dataset | Baseline – | Stage 1 CUB200 | Stage 2 PixmoCount | Stage 3 PathVQA | Stage 4 TextVQA | Stage 5 TimeClock |
|---|---|---|---|---|---|---|
| **Target** | | | | | | |
| CUB200 | 53.7 | 80.6 | 80.0 | 78.0 | 78.2 | 77.0 |
| PixmoCount | 52.4 | 52.6 | 67.8 | 64.2 | 62.5 | 63.7 |
| PathVQA | 36.3 | 36.2 | 35.1 | 58.1 | 51.6 | 53.3 |
| TextVQA | 76.0 | 76.0 | 75.2 | 75.2 | 79.1 | 76.2 |
| TimeClock | 1.1 | 1.0 | 1.0 | 1.0 | 1.2 | 33.7 |
| **Average** | 43.9 | 49.3 | 51.8 | 55.3 | 54.5 | 60.8 |
| **Held out** | | | | | | |
| AI2D | 81.4 | 81.9 | 80.8 | 79.7 | 81.3 | 79.7 |
| ChartQA | 80.1 | 80.1 | 79.2 | 78.1 | 79.1 | 71.7 |
| DocVQA | 87.1 | 87.1 | 85.0 | 83.4 | 84.4 | 74.1 |
| InfoVQA | 65.9 | 66.1 | 63.9 | 62.5 | 64.4 | 59.3 |
| MMStar | 61.8 | 62.1 | 62.2 | 60.5 | 61.2 | 60.3 |
| RealWorldQA | 66.4 | 67.6 | 66.4 | 65.9 | 68.8 | 66.3 |
| ScienceQA | 95.9 | 96.0 | 95.4 | 95.2 | 95.6 | 93.5 |
| SeedBench | 72.4 | 72.5 | 72.2 | 71.9 | 72.6 | 72.4 |
| **Average** | 76.4 | 76.7 | 75.6 | 74.7 | 75.9 | 72.2 |

through $E_{\text{pt}}$. During task $t$, we *freeze* $E_{\text{pt}}$ and update only the parameters of $E_{\text{new}}$ and the gate. Because $E_{\text{pt}}$ remains untouched, it acts as a safeguard when the tuned expert fails, giving MoE an inherent resistance to forgetting. We repeat this procedure for every new task, always reusing the same pair $(E_{\text{pt}}, E_{\text{new}})$ and thus adding no extra parameters beyond the current tuned expert and gate.

Table 9: Detailed performance of using **LwF** to mitigate forgetting by performing sequential fine-tuning on each target task.

| Dataset | Baseline – | Stage 1 CUB200 | Stage 2 PixmoCount | Stage 3 PathVQA | Stage 4 TextVQA | Stage 5 TimeClock |
|---|---|---|---|---|---|---|
| **Target** | | | | | | |
| CUB200 | 53.7 | 90.0 | 89.8 | 89.6 | 89.5 | 89.3 |
| PixmoCount | 52.4 | 53.2 | 67.6 | 67.8 | 61.6 | 61.0 |
| PathVQA | 36.3 | 35.9 | 35.0 | 61.1 | 58.3 | 57.6 |
| TextVQA | 76.0 | 76.3 | 76.5 | 76.6 | 80.3 | 79.4 |
| TimeClock | 1.1 | 1.0 | 1.7 | 1.0 | 1.4 | 67.8 |
| **Average** | 43.9 | 51.3 | 54.1 | 59.2 | 58.2 | 71.0 |
| **Held out** | | | | | | |
| AI2D | 81.4 | 81.9 | 81.6 | 81.6 | 81.6 | 81.7 |
| ChartQA | 80.1 | 79.9 | 79.9 | 79.9 | 80.2 | 78.0 |
| DocVQA | 87.1 | 87.1 | 86.6 | 86.5 | 86.5 | 85.6 |
| InfoVQA | 65.9 | 66.3 | 66.0 | 65.8 | 66.1 | 65.2 |
| MMStar | 61.8 | 62.4 | 63.2 | 62.6 | 62.0 | 61.6 |
| RealWorldQA | 66.4 | 67.8 | 59.6 | 64.4 | 67.8 | 66.9 |
| ScienceQA | 95.9 | 96.0 | 93.0 | 95.8 | 96.3 | 96.0 |
| SeedBench | 72.4 | 72.4 | 72.5 | 72.2 | 72.5 | 72.5 |
| **Average** | 76.4 | 76.7 | 75.3 | 76.1 | 76.6 | 75.9 |

Table 10: Detailed performance of using **WiSE-FT** using $\beta = 0.3$ to mitigate forgetting by performing sequential fine-tuning on each target task.

| Dataset | Baseline – | Stage 1 CUB200 | Stage 2 PixmoCount | Stage 3 PathVQA | Stage 4 TextVQA | Stage 5 TimeClock |
|---|---|---|---|---|---|---|
| **Target** | | | | | | |
| CUB200 | 53.7 | 89.9 | 89.2 | 89.5 | 89.0 | 89.0 |
| PixmoCount | 52.4 | 53.7 | 69.7 | 68.0 | 68.0 | 66.1 |
| PathVQA | 36.3 | 35.9 | 34.2 | 59.1 | 56.3 | 56.1 |
| TextVQA | 76.0 | 76.4 | 74.2 | 76.1 | 79.7 | 78.3 |
| TimeClock | 1.1 | 1.0 | 1.8 | 1.0 | 1.4 | 62.2 |
| **Average** | 43.9 | 51.4 | 53.8 | 58.7 | 58.9 | 70.3 |
| **Held out** | | | | | | |
| AI2D | 81.4 | 81.8 | 81.8 | 81.6 | 81.6 | 81.8 |
| ChartQA | 80.1 | 80.1 | 80.4 | 80.2 | 80.2 | 78.3 |
| DocVQA | 87.1 | 87.3 | 85.2 | 86.7 | 86.4 | 85.1 |
| InfoVQA | 65.9 | 65.7 | 64.0 | 65.3 | 65.9 | 64.1 |
| MMStar | 61.8 | 61.9 | 61.4 | 61.6 | 62.3 | 60.9 |
| RealWorldQA | 66.4 | 67.7 | 63.9 | 68.4 | 69.7 | 67.8 |
| ScienceQA | 95.9 | 96.3 | 96.3 | 96.3 | 96.4 | 96.3 |
| SeedBench | 72.4 | 72.6 | 72.9 | 72.6 | 72.6 | 72.6 |
| **Average** | 76.4 | 76.7 | 75.7 | 76.6 | 76.9 | 75.9 |

# H    DETAILED TASK PERFORMANCE

## H.1    FORGETTING MITIGATION METHODS SEQUENTIAL TUNING TABLES

Tab. 8, Tab. 9, Tab. 10, and Tab. 7 are detailed sequential tuning tables for forgetting mitigation approaches tested in the paper, i.e., LoRA, LwF (Li & Hoiem, 2017), WiSE-FT (Wortsman et al., 2022), and MoE.

Table 11: Detailed performance of sequentially fine-tuning the **full** model on each target task.

| Dataset | Baseline – | Stage 1 CUB200 | Stage 2 PixmoCount | Stage 3 PathVQA | Stage 4 TextVQA | Stage 5 TimeClock |
|---|---|---|---|---|---|---|
| **Target** | | | | | | |
| CUB200 | 53.7 | 90.7 | 89.2 | 88.1 | 88.0 | 86.9 |
| PixmoCount | 52.4 | 54.9 | 73.0 | 64.6 | 63.1 | 59.4 |
| PathVQA | 36.3 | 34.8 | 3.7 | 63.6 | 59.8 | 58.6 |
| TextVQA | 76.0 | 76.6 | 59.0 | 74.6 | 79.6 | 68.9 |
| TimeClock | 1.1 | 1.0 | 1.4 | 1.2 | 1.5 | 46.9 |
| **Average** | 43.9 | 51.6 | 45.3 | 58.4 | 58.4 | 64.1 |
| **Held out** | | | | | | |
| AI2D | 81.4 | 81.4 | 57.9 | 80.4 | 80.3 | 74.7 |
| ChartQA | 80.1 | 80.3 | 63.8 | 77.9 | 77.6 | 66.9 |
| DocVQA | 87.1 | 87.4 | 74.1 | 83.1 | 82.9 | 68.6 |
| InfoVQA | 65.9 | 65.7 | 54.2 | 62.5 | 61.9 | 50.3 |
| MMStar | 61.8 | 60.6 | 59.6 | 58.9 | 59.0 | 53.9 |
| RealWorldQA | 66.4 | 68.6 | 44.2 | 63.4 | 66.1 | 55.8 |
| ScienceQA | 95.9 | 95.0 | 76.0 | 94.6 | 93.5 | 87.9 |
| SeedBench | 72.4 | 72.6 | 71.7 | 70.3 | 71.6 | 65.9 |
| **Average** | 76.4 | 76.5 | 62.7 | 73.9 | 74.1 | 65.5 |

Table 12: Detailed performance of sequentially fine-tuning the **vision tower** on each target task.

| Dataset | Baseline – | Stage 1 CUB200 | Stage 2 PixmoCount | Stage 3 PathVQA | Stage 4 TextVQA | Stage 5 TimeClock |
|---|---|---|---|---|---|---|
| **Target** | | | | | | |
| CUB200 | 53.7 | 69.9 | 57.5 | 58.8 | 61.5 | 55.3 |
| PixmoCount | 52.4 | 22.1 | 64.2 | 44.0 | 59.0 | 37.8 |
| PathVQA | 36.3 | 35.4 | 31.8 | 37.0 | 34.4 | 34.2 |
| TextVQA | 76.0 | 75.5 | 72.2 | 70.7 | 76.4 | 72.7 |
| TimeClock | 1.1 | 1.0 | 1.0 | 0.8 | 1.1 | 26.3 |
| **Average** | 43.9 | 40.8 | 45.3 | 42.3 | 46.5 | 45.3 |
| **Held out** | | | | | | |
| AI2D | 81.4 | 81.3 | 79.7 | 78.9 | 80.6 | 77.3 |
| ChartQA | 80.1 | 80.0 | 76.9 | 74.3 | 79.5 | 76.0 |
| DocVQA | 87.1 | 85.8 | 79.6 | 75.1 | 85.1 | 78.9 |
| InfoVQA | 65.9 | 63.6 | 60.3 | 56.3 | 64.4 | 59.6 |
| MMStar | 61.8 | 61.4 | 57.6 | 56.4 | 59.5 | 55.6 |
| RealWorldQA | 66.4 | 66.3 | 65.0 | 61.3 | 65.6 | 61.8 |
| ScienceQA | 95.9 | 94.6 | 91.9 | 90.5 | 94.3 | 89.4 |
| SeedBench | 72.4 | 71.5 | 70.6 | 69.6 | 71.0 | 69.5 |
| **Average** | 76.4 | 75.6 | 72.7 | 70.3 | 75.0 | 71.0 |

## H.2 SEQUENTIAL TUNING DETAILED PERFORMANCE TABLES ON LLAVA-ONEVISION

We include the detailed task performances for sequential fine-tuning experiments on LLaVA-OneVision here. Tab. 11, Tab. 12, Tab. 13, Tab. 14, Tab. 15, Tab. 16, Tab. 17, and Tab. 18 are detailed performance tables of sequentially fine-tuning the Full model, Vision Encoder, Projector, LLM, SA projection layers in LLM, SA Proj. (QKV), MLP layers in LLM, MLP (Gate&Up), respectively.

Table 13: Detailed performance of sequentially fine-tuning the **projector** in the LLM on each target task.

| Dataset | Baseline – | Stage 1 CUB200 | Stage 2 PixmoCount | Stage 3 PathVQA | Stage 4 TextVQA | Stage 5 TimeClock |
|---|---|---|---|---|---|---|
| **Target** | | | | | | |
| CUB200 | 53.7 | 59.4 | 57.7 | 57.5 | 58.2 | 57.4 |
| PixmoCount | 52.4 | 53.2 | 56.6 | 57.9 | 58.2 | 57.7 |
| PathVQA | 36.3 | 36.1 | 36.0 | 35.5 | 36.4 | 35.9 |
| TextVQA | 76.0 | 76.1 | 76.3 | 76.4 | 77.0 | 76.9 |
| TimeClock | 1.1 | 1.1 | 1.2 | 1.2 | 1.0 | 1.9 |
| **Average** | 43.9 | 45.2 | 45.6 | 45.7 | 46.2 | 46.0 |
| **Held out** | | | | | | |
| AI2D | 81.4 | 81.4 | 81.7 | 81.6 | 81.8 | 81.1 |
| ChartQA | 80.1 | 79.9 | 80.2 | 80.0 | 80.1 | 79.4 |
| DocVQA | 87.1 | 87.3 | 87.2 | 86.1 | 86.3 | 86.2 |
| InfoVQA | 65.9 | 66.1 | 66.1 | 65.5 | 66.3 | 65.1 |
| MMStar | 61.8 | 62.1 | 61.7 | 60.9 | 61.0 | 60.5 |
| RealWorldQA | 66.4 | 66.1 | 66.9 | 67.3 | 68.0 | 67.1 |
| ScienceQA | 95.9 | 95.9 | 96.0 | 95.9 | 95.8 | 95.6 |
| SeedBench | 72.4 | 72.6 | 72.5 | 72.3 | 72.5 | 72.4 |
| **Average** | 76.4 | 76.4 | 76.5 | 76.2 | 76.5 | 75.9 |

Table 14: Detailed performance of sequentially fine-tuning the **LLM** on each target task.

| Dataset | Baseline – | Stage 1 CUB200 | Stage 2 PixmoCount | Stage 3 PathVQA | Stage 4 TextVQA | Stage 5 TimeClock |
|---|---|---|---|---|---|---|
| **Target** | | | | | | |
| CUB200 | 53.7 | 90.7 | 89.4 | 89.6 | 88.9 | 87.8 |
| PixmoCount | 52.4 | 54.3 | 70.2 | 62.2 | 63.3 | 56.6 |
| PathVQA | 36.3 | 35.2 | 4.4 | 63.2 | 58.6 | 56.7 |
| TextVQA | 76.0 | 76.6 | 60.5 | 74.7 | 79.6 | 71.2 |
| TimeClock | 1.1 | 1.0 | 1.4 | 1.0 | 1.3 | 71.8 |
| **Average** | 43.9 | 51.6 | 45.2 | 58.1 | 58.3 | 68.8 |
| **Held out** | | | | | | |
| AI2D | 81.4 | 81.2 | 72.8 | 80.7 | 79.8 | 75.2 |
| ChartQA | 80.1 | 80.4 | 66.9 | 78.4 | 78.0 | 68.6 |
| DocVQA | 87.1 | 87.3 | 75.9 | 84.2 | 83.0 | 72.0 |
| InfoVQA | 65.9 | 65.8 | 54.9 | 62.8 | 61.6 | 51.8 |
| MMStar | 61.8 | 60.8 | 58.4 | 58.9 | 59.2 | 53.4 |
| RealWorldQA | 66.4 | 67.5 | 46.4 | 63.5 | 67.2 | 59.0 |
| ScienceQA | 95.9 | 94.8 | 83.6 | 94.7 | 92.5 | 90.0 |
| SeedBench | 72.4 | 72.7 | 72.3 | 71.0 | 71.5 | 65.3 |
| **Average** | 76.4 | 76.3 | 66.4 | 74.3 | 74.1 | 66.9 |

## H.3 SEQUENTIAL TUNING DETAILED PERFORMANCE TABLES ON LLAVA-NEXT (LLAMA 3)

We include the detailed task performances for sequential fine-tuning experiments on LLaVA-NeXT (LLaMA 3) here. Tab. 19, Tab. 20, Tab. 21, Tab. 22, Tab. 23, and Tab. 24 are detailed performance tables of sequentially fine-tuning the Full model, Vision Encoder + Projector, LLM, SA projection layers in LLM, MLP layers in LLM, and MLP (Gate&Up), respectively.

Table 15: Detailed performance of sequentially fine-tuning the **SA projection layers in the LLM** on each target task.

| Dataset | Baseline – | Stage 1 CUB200 | Stage 2 PixmoCount | Stage 3 PathVQA | Stage 4 TextVQA | Stage 5 TimeClock |
|---|---|---|---|---|---|---|
| **Target** | | | | | | |
| CUB200 | 53.7 | 85.5 | 85.1 | 84.8 | 84.4 | 84.0 |
| PixmoCount | 52.4 | 53.9 | 67.8 | 68.2 | 64.8 | 66.3 |
| PathVQA | 36.3 | 35.7 | 35.0 | 55.9 | 52.6 | 51.4 |
| TextVQA | 76.0 | 76.1 | 76.4 | 75.8 | 79.3 | 78.9 |
| TimeClock | 1.1 | 1.0 | 1.2 | 1.0 | 1.2 | 52.6 |
| **Average** | 43.9 | 50.4 | 53.1 | 57.1 | 56.5 | 66.6 |
| **Held out** | | | | | | |
| AI2D | 81.4 | 82.0 | 81.4 | 81.2 | 81.9 | 81.9 |
| ChartQA | 80.1 | 80.0 | 79.7 | 80.0 | 80.6 | 79.4 |
| DocVQA | 87.1 | 87.2 | 86.9 | 86.8 | 86.3 | 86.1 |
| InfoVQA | 65.9 | 66.0 | 64.7 | 65.3 | 66.0 | 64.9 |
| MMStar | 61.8 | 62.4 | 62.3 | 62.1 | 62.4 | 61.9 |
| RealWorldQA | 66.4 | 68.0 | 67.1 | 66.9 | 69.2 | 68.9 |
| ScienceQA | 95.9 | 95.7 | 95.9 | 96.1 | 96.3 | 96.1 |
| SeedBench | 72.4 | 72.3 | 72.4 | 72.0 | 72.5 | 72.4 |
| **Average** | 76.4 | 76.7 | 76.3 | 76.3 | 76.9 | 76.5 |

Table 16: Detailed performance of sequentially fine-tuning the **SA Proj. (QKV)** in the LLM on each target task.

| Dataset | Baseline – | Stage 1 CUB200 | Stage 2 PixmoCount | Stage 3 PathVQA | Stage 4 TextVQA | Stage 5 TimeClock |
|---|---|---|---|---|---|---|
| **Target** | | | | | | |
| CUB200 | 53.7 | 78.7 | 78.4 | 78.4 | 78.2 | 77.6 |
| PixmoCount | 52.4 | 53.2 | 65.9 | 67.2 | 62.4 | 65.4 |
| PathVQA | 36.3 | 36.1 | 36.4 | 44.2 | 42.6 | 43.0 |
| TextVQA | 76.0 | 76.2 | 76.9 | 76.2 | 78.6 | 78.3 |
| TimeClock | 1.1 | 1.0 | 1.0 | 0.8 | 1.1 | 32.2 |
| **Average** | 43.9 | 49.0 | 51.7 | 53.4 | 52.6 | 59.3 |
| **Held out** | | | | | | |
| AI2D | 81.4 | 81.9 | 81.7 | 81.2 | 82.0 | 81.9 |
| ChartQA | 80.1 | 79.9 | 79.9 | 79.8 | 80.4 | 79.9 |
| DocVQA | 87.1 | 87.2 | 87.3 | 86.9 | 86.7 | 86.4 |
| InfoVQA | 65.9 | 65.8 | 65.4 | 65.7 | 65.9 | 65.8 |
| MMStar | 61.8 | 62.3 | 62.5 | 62.7 | 62.3 | 62.3 |
| RealWorldQA | 66.4 | 67.6 | 67.6 | 67.1 | 68.4 | 68.8 |
| ScienceQA | 95.9 | 95.9 | 95.8 | 95.9 | 96.3 | 96.1 |
| SeedBench | 72.4 | 72.2 | 72.4 | 72.1 | 72.3 | 72.3 |
| **Average** | 76.4 | 76.6 | 76.6 | 76.4 | 76.8 | 76.7 |

## H.4 SEQUENTIAL TUNING DETAILED PERFORMANCE TABLES ON QWEN2.5-VL

We include the detailed task performances for sequential fine-tuning experiments on Qwen2.5-VL here. Tab. 19, Tab. 20, Tab. 21, Tab. 22, Tab. 23, and Tab. 24 are detailed performance tables of sequentially fine-tuning the Full model, Vision Encoder + Projector, LLM, SA projection layers in LLM, MLP layers in LLM, and MLP (Gate&Up), respectively.

Table 17: Detailed performance of sequentially fine-tuning the **MLP in the LLM** on each target task.

| Dataset | Baseline
– | Stage 1
CUB200 | Stage 2
PixmoCount | Stage 3
PathVQA | Stage 4
TextVQA | Stage 5
TimeClock |
|---|---|---|---|---|---|---|
| **Target** | | | | | | |
| CUB200 | 53.7 | 90.1 | 89.5 | 89.6 | 89.3 | 88.9 |
| PixmoCount | 52.4 | 54.1 | 71.5 | 67.6 | 68.0 | 62.0 |
| PathVQA | 36.3 | 35.6 | 17.0 | 64.1 | 60.9 | 60.9 |
| TextVQA | 76.0 | 76.6 | 66.2 | 75.3 | 79.8 | 74.0 |
| TimeClock | 1.1 | 1.0 | 1.5 | 1.2 | 1.6 | 74.0 |
| **Average** | 43.9 | 51.5 | 49.1 | 59.6 | 59.9 | 72.0 |
| **Held out** | | | | | | |
| AI2D | 81.4 | 81.7 | 80.9 | 81.0 | 80.5 | 80.4 |
| ChartQA | 80.1 | 80.4 | 75.7 | 79.7 | 79.9 | 75.1 |
| DocVQA | 87.1 | 87.2 | 80.0 | 85.5 | 84.5 | 78.9 |
| InfoVQA | 65.9 | 65.8 | 60.0 | 64.0 | 63.7 | 59.3 |
| MMStar | 61.8 | 61.5 | 61.2 | 61.0 | 60.7 | 59.5 |
| RealWorldQA | 66.4 | 68.0 | 53.7 | 65.9 | 68.1 | 62.1 |
| ScienceQA | 95.9 | 96.1 | 93.7 | 96.2 | 95.9 | 95.2 |
| SeedBench | 72.4 | 72.7 | 72.7 | 72.1 | 72.3 | 71.9 |
| **Average** | 76.4 | 76.7 | 72.2 | 75.7 | 75.7 | 72.8 |

Table 18: Detailed performance of sequentially fine-tuning the **MLP (Gate & Up)** in the LLM on each target task.

| Dataset | Baseline
– | Stage 1
CUB200 | Stage 2
PixmoCount | Stage 3
PathVQA | Stage 4
TextVQA | Stage 5
TimeClock |
|---|---|---|---|---|---|---|
| **Target** | | | | | | |
| CUB200 | 53.7 | 90.2 | 89.8 | 89.8 | 89.6 | 89.5 |
| PixmoCount | 52.4 | 53.4 | 71.5 | 67.8 | 68.4 | 67.2 |
| PathVQA | 36.3 | 36.1 | 35.0 | 61.9 | 58.5 | 58.9 |
| TextVQA | 76.0 | 76.4 | 75.5 | 76.1 | 80.0 | 79.3 |
| TimeClock | 1.1 | 0.9 | 1.8 | 1.2 | 1.9 | 72.2 |
| **Average** | 43.9 | 51.4 | 54.7 | 59.4 | 59.7 | 73.4 |
| **Held out** | | | | | | |
| AI2D | 81.4 | 81.7 | 81.6 | 81.3 | 81.2 | 81.5 |
| ChartQA | 80.1 | 80.1 | 80.6 | 80.1 | 80.7 | 78.8 |
| DocVQA | 87.1 | 87.0 | 86.3 | 86.6 | 85.9 | 85.4 |
| InfoVQA | 65.9 | 66.1 | 65.4 | 65.0 | 65.3 | 64.9 |
| MMStar | 61.8 | 62.2 | 63.1 | 62.7 | 62.5 | 62.0 |
| RealWorldQA | 66.4 | 67.7 | 64.4 | 67.8 | 69.5 | 68.4 |
| ScienceQA | 95.9 | 96.3 | 96.4 | 96.5 | 96.2 | 96.0 |
| SeedBench | 72.4 | 72.4 | 72.6 | 72.3 | 72.6 | 72.5 |
| **Average** | 76.4 | 76.7 | 76.3 | 76.5 | 76.7 | 76.2 |

Table 19: Detailed performance of sequentially fine-tuning the **full model** of **LLaVA-NeXT (LLaMA 3)** on each target task.

| Dataset | Baseline
– | Stage 1
CUB200 | Stage 2
PixmoCount | Stage 3
PathVQA | Stage 4
TextVQA | Stage 5
TimeClock |
|---|---|---|---|---|---|---|
| **Target** | | | | | | |
| CUB200 | 32.6 | 84.8 | 76.8 | 77.2 | 76.6 | 69.6 |
| PixmoCount | 45.7 | 37.6 | 63.3 | 48.5 | 32.4 | 44.0 |
| PathVQA | 13.2 | 24.8 | 0.7 | 62.0 | 55.6 | 45.2 |
| TextVQA | 65.4 | 52.2 | 31.0 | 56.1 | 72.9 | 42.8 |
| TimeClock | 0.8 | 0.3 | 0.1 | 0.6 | 0.5 | 33.1 |
| **Average** | 31.5 | 39.9 | 34.4 | 48.9 | 47.6 | 46.9 |
| **Held out** | | | | | | |
| AI2D | 71.6 | 54.0 | 53.3 | 62.1 | 58.2 | 43.9 |
| ChartQA | 69.2 | 54.3 | 14.6 | 48.6 | 51.0 | 7.8 |
| DocVQA | 72.7 | 40.4 | 27.7 | 46.6 | 59.2 | 15.7 |
| InfoVQA | 31.9 | 23.4 | 14.6 | 27.2 | 33.9 | 10.2 |
| MMStar | 42.0 | 43.9 | 41.5 | 39.6 | 42.4 | 25.6 |
| RealWorldQA | 59.7 | 55.3 | 32.7 | 50.3 | 53.6 | 19.2 |
| ScienceQA | 73.2 | 63.3 | 57.5 | 69.7 | 66.4 | 58.3 |
| SeedBench | 58.5 | 56.8 | 55.8 | 53.9 | 56.6 | 42.0 |
| **Average** | 59.8 | 48.9 | 37.2 | 49.7 | 52.7 | 27.8 |

Table 20: Detailed performance of sequentially fine-tuning the **vision encoder and projector** of **LLaVA-NeXT (LLaMA 3)** on each target task.

| Dataset | Baseline
– | Stage 1
CUB200 | Stage 2
PixmoCount | Stage 3
PathVQA | Stage 4
TextVQA | Stage 5
TimeClock |
|---|---|---|---|---|---|---|
| **Target** | | | | | | |
| CUB200 | 32.6 | 4.2 | 4.2 | 12.8 | 22.1 | 13.4 |
| PixmoCount | 45.7 | 0.6 | 44.9 | 40.8 | 48.9 | 40.3 |
| PathVQA | 13.2 | 0.3 | 1.8 | 34.1 | 34.1 | 33.1 |
| TextVQA | 65.4 | 0.9 | 0.9 | 50.1 | 69.9 | 59.3 |
| TimeClock | 0.8 | 0.0 | 0.6 | 0.5 | 0.7 | 5.2 |
| **Average** | 31.5 | 1.2 | 10.5 | 27.7 | 35.1 | 30.3 |
| **Held out** | | | | | | |
| AI2D | 71.6 | 13.8 | 5.8 | 57.1 | 64.9 | 58.0 |
| ChartQA | 69.2 | 0.1 | 0.5 | 34.9 | 55.3 | 42.7 |
| DocVQA | 72.7 | 0.6 | 1.4 | 36.3 | 59.2 | 38.9 |
| InfoVQA | 31.9 | 0.3 | 0.2 | 22.2 | 29.9 | 24.4 |
| MMStar | 42.0 | 9.9 | 1.9 | 38.1 | 42.3 | 35.8 |
| RealWorldQA | 59.7 | 14.4 | 2.4 | 55.6 | 59.7 | 52.3 |
| ScienceQA | 73.2 | 11.3 | 0.0 | 64.0 | 69.9 | 67.3 |
| SeedBench | 58.5 | 15.1 | 6.0 | 49.7 | 58.7 | 52.3 |
| **Average** | 59.8 | 8.2 | 2.3 | 44.7 | 55.0 | 46.5 |

Table 21: Detailed performance of sequentially fine-tuning the **LLM** of **LLaVA-NeXT (LLaMA 3)** on each target task.

| Dataset | Baseline – | Stage 1 CUB200 | Stage 2 PixmoCount | Stage 3 PathVQA | Stage 4 TextVQA | Stage 5 TimeClock |
|---|---|---|---|---|---|---|
| **Target** | | | | | | |
| CUB200 | 32.6 | 85.2 | 68.6 | 72.0 | 72.1 | 68.8 |
| PixmoCount | 45.7 | 32.0 | 57.5 | 55.1 | 21.9 | 41.6 |
| PathVQA | 13.2 | 23.3 | 14.1 | 62.7 | 56.4 | 42.7 |
| TextVQA | 65.4 | 56.9 | 35.0 | 57.8 | 72.6 | 40.1 |
| TimeClock | 0.8 | 0.1 | 0.0 | 0.8 | 0.6 | 60.9 |
| **Average** | 31.5 | 39.5 | 35.0 | 49.7 | 44.7 | 50.8 |
| **Held out** | | | | | | |
| AI2D | 71.6 | 55.5 | 54.4 | 62.7 | 59.9 | 35.1 |
| ChartQA | 69.2 | 54.3 | 20.0 | 49.4 | 53.7 | 4.3 |
| DocVQA | 72.7 | 45.7 | 31.6 | 49.5 | 58.6 | 21.5 |
| InfoVQA | 31.9 | 25.2 | 11.5 | 27.4 | 33.4 | 8.7 |
| MMStar | 42.0 | 42.6 | 40.8 | 40.8 | 38.4 | 16.3 |
| RealWorldQA | 59.7 | 56.1 | 11.8 | 50.8 | 57.1 | 16.1 |
| ScienceQA | 73.2 | 65.5 | 24.3 | 70.8 | 68.7 | 52.2 |
| SeedBench | 58.5 | 56.8 | 55.4 | 54.7 | 56.8 | 37.2 |
| **Average** | 59.8 | 50.2 | 31.2 | 50.8 | 53.3 | 23.9 |

Table 22: Detailed performance of sequentially fine-tuning the **SA projection layers in the LLM** of **LLaVA-NeXT (LLaMA 3)** on each target task.

| Dataset | Baseline – | Stage 1 CUB200 | Stage 2 PixmoCount | Stage 3 PathVQA | Stage 4 TextVQA | Stage 5 TimeClock |
|---|---|---|---|---|---|---|
| **Target** | | | | | | |
| CUB200 | 32.6 | 78.1 | 68.2 | 72.3 | 70.6 | 70.3 |
| PixmoCount | 45.7 | 38.8 | 60.5 | 50.6 | 9.9 | 54.7 |
| PathVQA | 13.2 | 28.7 | 28.4 | 53.9 | 43.9 | 44.3 |
| TextVQA | 65.4 | 64.3 | 62.9 | 62.0 | 73.5 | 64.9 |
| TimeClock | 0.8 | 0.7 | 0.8 | 0.8 | 0.6 | 32.9 |
| **Average** | 31.5 | 42.1 | 44.2 | 47.9 | 39.7 | 53.4 |
| **Held out** | | | | | | |
| AI2D | 71.6 | 67.6 | 65.3 | 68.5 | 67.4 | 65.4 |
| ChartQA | 69.2 | 60.4 | 58.3 | 61.3 | 63.8 | 49.4 |
| DocVQA | 72.7 | 60.8 | 58.9 | 58.3 | 63.6 | 48.9 |
| InfoVQA | 31.9 | 29.5 | 32.1 | 33.9 | 35.8 | 27.5 |
| MMStar | 42.0 | 46.9 | 45.2 | 43.1 | 42.5 | 41.1 |
| RealWorldQA | 59.7 | 58.6 | 55.4 | 58.3 | 63.3 | 53.9 |
| ScienceQA | 73.2 | 72.2 | 70.4 | 73.4 | 72.6 | 70.7 |
| SeedBench | 58.5 | 59.6 | 60.1 | 58.2 | 59.9 | 60.2 |
| **Average** | 59.8 | 57.0 | 55.7 | 56.9 | 58.6 | 52.1 |

Table 23: Detailed performance of sequentially fine-tuning the **MLP in the LLM** of **LLaVA-NeXT (LLaMA 3)** on each target task.

| Dataset | Baseline – | Stage 1 CUB200 | Stage 2 PixmoCount | Stage 3 PathVQA | Stage 4 TextVQA | Stage 5 TimeClock |
|---|---|---|---|---|---|---|
| **Target** | | | | | | |
| CUB200 | 32.6 | 84.3 | 78.7 | 76.9 | 76.9 | 72.0 |
| PixmoCount | 45.7 | 34.6 | 58.6 | 53.9 | 35.8 | 52.8 |
| PathVQA | 13.2 | 28.5 | 26.3 | 61.8 | 56.2 | 52.7 |
| TextVQA | 65.4 | 61.5 | 54.6 | 62.0 | 73.2 | 59.2 |
| TimeClock | 0.8 | 0.5 | 0.8 | 0.8 | 0.7 | 54.2 |
| **Average** | 31.5 | 41.9 | 43.8 | 51.1 | 48.6 | 58.2 |
| **Held out** | | | | | | |
| AI2D | 71.6 | 65.3 | 62.8 | 66.3 | 62.1 | 59.1 |
| ChartQA | 69.2 | 59.8 | 50.4 | 58.8 | 57.6 | 32.2 |
| DocVQA | 72.7 | 54.1 | 48.6 | 57.0 | 61.2 | 38.8 |
| InfoVQA | 31.9 | 29.7 | 24.3 | 32.5 | 35.3 | 22.4 |
| MMStar | 42.0 | 44.9 | 43.7 | 43.5 | 41.2 | 35.2 |
| RealWorldQA | 59.7 | 58.3 | 50.3 | 53.5 | 57.3 | 39.5 |
| ScienceQA | 73.2 | 71.7 | 69.8 | 70.3 | 62.8 | 65.6 |
| SeedBench | 58.5 | 59.5 | 58.5 | 57.6 | 58.8 | 55.7 |
| **Average** | 59.8 | 55.4 | 51.1 | 54.9 | 54.5 | 43.6 |

Table 24: Detailed performance of sequentially fine-tuning the **MLP (Gate & Up) in the LLM** of **LLaVA-NeXT (LLaMA 3)** on each target task.

| Dataset | Baseline – | Stage 1 CUB200 | Stage 2 PixmoCount | Stage 3 PathVQA | Stage 4 TextVQA | Stage 5 TimeClock |
|---|---|---|---|---|---|---|
| **Target** | | | | | | |
| CUB200 | 32.6 | 78.4 | 73.9 | 74.3 | 71.5 | 71.9 |
| PixmoCount | 45.7 | 32.4 | 58.4 | 57.9 | 29.8 | 44.8 |
| PathVQA | 13.2 | 27.9 | 27.9 | 59.5 | 51.6 | 54.0 |
| TextVQA | 65.4 | 63.5 | 63.7 | 64.7 | 74.0 | 63.9 |
| TimeClock | 0.8 | 0.8 | 0.6 | 0.7 | 0.8 | 27.6 |
| **Average** | 31.5 | 40.6 | 44.9 | 51.4 | 45.5 | 52.4 |
| **Held out** | | | | | | |
| AI2D | 71.6 | 67.6 | 67.4 | 68.9 | 66.7 | 66.1 |
| ChartQA | 69.2 | 60.6 | 63.3 | 65.3 | 64.0 | 46.6 |
| DocVQA | 72.7 | 59.7 | 62.4 | 63.8 | 64.1 | 47.6 |
| InfoVQA | 31.9 | 32.3 | 33.0 | 36.2 | 36.5 | 27.5 |
| MMStar | 42.0 | 45.8 | 45.5 | 46.0 | 43.6 | 40.9 |
| RealWorldQA | 59.7 | 60.4 | 52.4 | 54.9 | 59.0 | 49.2 |
| ScienceQA | 73.2 | 72.3 | 71.7 | 73.1 | 71.8 | 71.9 |
| SeedBench | 58.5 | 60.1 | 59.9 | 59.0 | 60.0 | 59.2 |
| **Average** | 59.8 | 57.4 | 56.9 | 58.4 | 58.2 | 51.1 |

Table 25: Detailed performance of sequentially fine-tuning the **full model** of **Qwen2.5-VL** on each target task.

| Dataset | Baseline – | Stage 1 CUB200 | Stage 2 PixmoCount | Stage 3 PathVQA | Stage 4 TextVQA | Stage 5 TimeClock |
|---|---|---|---|---|---|---|
| **Target** | | | | | | |
| CUB200 | 81.4 | 93.5 | 0.2 | 12.1 | 92.6 | 92.4 |
| PixmoCount | 58.6 | 55.6 | 50.6 | 48.3 | 47.4 | 51.1 |
| PathVQA | 29.2 | 18.5 | 0.0 | 60.8 | 58.2 | 59.1 |
| TextVQA | 83.0 | 69.5 | 17.2 | 73.3 | 81.5 | 62.7 |
| TimeClock | 8.2 | 0.1 | 0.0 | 0.0 | 6.3 | 60.8 |
| **Average** | 52.1 | 47.4 | 13.6 | 38.9 | 57.2 | 65.2 |
| **Held out** | | | | | | |
| AI2D | 82.9 | 79.5 | 0.1 | 64.0 | 78.8 | 72.5 |
| ChartQA | 83.2 | 72.6 | 54.2 | 72.1 | 69.3 | 62.7 |
| DocVQA | 94.4 | 77.2 | 30.2 | 76.1 | 90.0 | 66.8 |
| InfoVQA | 80.3 | 61.9 | 33.6 | 64.8 | 74.5 | 47.1 |
| MMStar | 62.6 | 59.3 | 0.0 | 34.0 | 53.5 | 46.9 |
| RealWorldQA | 68.6 | 59.5 | 3.7 | 27.5 | 59.7 | 51.5 |
| ScienceQA | 76.7 | 77.8 | 0.4 | 43.4 | 77.3 | 71.6 |
| SeedBench | 74.1 | 72.0 | 0.0 | 24.6 | 68.9 | 63.7 |
| **Average** | 77.9 | 70.0 | 15.3 | 50.8 | 71.5 | 60.4 |

Table 26: Detailed performance of sequentially fine-tuning the **vision encoder and projector** of **Qwen2.5-VL** on each target task.

| Dataset | Baseline – | Stage 1 CUB200 | Stage 2 PixmoCount | Stage 3 PathVQA | Stage 4 TextVQA | Stage 5 TimeClock |
|---|---|---|---|---|---|---|
| **Target** | | | | | | |
| CUB200 | 81.4 | 92.3 | 81.4 | 81.5 | 81.5 | 88.0 |
| PixmoCount | 58.6 | 56.4 | 59.0 | 59.2 | 58.2 | 33.1 |
| PathVQA | 29.2 | 30.3 | 29.1 | 29.3 | 30.2 | 35.4 |
| TextVQA | 83.0 | 82.5 | 83.2 | 83.1 | 83.1 | 71.0 |
| TimeClock | 8.2 | 8.5 | 8.4 | 8.6 | 8.8 | 57.5 |
| **Average** | 52.1 | 54.0 | 52.2 | 52.3 | 52.4 | 57.0 |
| **Held out** | | | | | | |
| AI2D | 82.9 | 83.0 | 82.8 | 82.9 | 83.1 | 75.4 |
| ChartQA | 83.2 | 83.8 | 83.7 | 83.8 | 83.9 | 75.1 |
| DocVQA | 94.4 | 94.4 | 94.5 | 94.4 | 94.5 | 88.7 |
| InfoVQA | 80.3 | 79.5 | 80.1 | 80.2 | 80.3 | 69.2 |
| MMStar | 62.6 | 62.3 | 62.5 | 62.9 | 63.4 | 52.7 |
| RealWorldQA | 68.6 | 67.6 | 68.5 | 68.5 | 69.9 | 62.0 |
| ScienceQA | 76.7 | 76.6 | 76.4 | 76.1 | 76.2 | 82.3 |
| SeedBench | 74.1 | 73.7 | 74.0 | 74.1 | 74.1 | 67.8 |
| **Average** | 77.9 | 77.6 | 77.8 | 77.9 | 78.2 | 71.6 |

Table 27: Detailed performance of sequentially fine-tuning the **LLM** of **Qwen2.5-VL** on each target task.

| Dataset | Baseline – | Stage 1 CUB200 | Stage 2 PixmoCount | Stage 3 PathVQA | Stage 4 TextVQA | Stage 5 TimeClock |
|---|---|---|---|---|---|---|
| **Target** | | | | | | |
| CUB200 | 81.4 | 93.8 | 0.0 | 64.4 | 67.6 | 91.5 |
| PixmoCount | 58.6 | 55.8 | 47.0 | 50.6 | 41.0 | 49.1 |
| PathVQA | 29.2 | 4.9 | 0.0 | 63.0 | 59.7 | 60.2 |
| TextVQA | 83.0 | 47.9 | 11.6 | 73.3 | 82.1 | 61.8 |
| TimeClock | 8.2 | 0.0 | 0.0 | 0.0 | 4.6 | 58.5 |
| **Average** | 52.1 | 40.5 | 11.7 | 50.3 | 51.0 | 64.2 |
| **Held out** | | | | | | |
| AI2D | 82.9 | 77.4 | 0.0 | 35.8 | 75.6 | 56.9 |
| ChartQA | 83.2 | 41.9 | 49.1 | 67.0 | 65.1 | 68.2 |
| DocVQA | 94.4 | 49.6 | 23.4 | 76.3 | 89.9 | 65.8 |
| InfoVQA | 80.3 | 41.6 | 28.5 | 60.8 | 74.5 | 49.3 |
| MMStar | 62.6 | 59.7 | 0.0 | 33.9 | 52.4 | 36.1 |
| RealWorldQA | 68.6 | 56.7 | 3.4 | 25.4 | 51.8 | 38.3 |
| ScienceQA | 76.7 | 77.6 | 0.0 | 39.5 | 69.9 | 59.3 |
| SeedBench | 74.1 | 71.7 | 0.0 | 21.9 | 61.8 | 51.8 |
| **Average** | 77.9 | 59.5 | 13.1 | 45.1 | 67.6 | 53.2 |

Table 28: Detailed performance of sequentially fine-tuning the **SA projection layers in the LLM** of **Qwen2.5-VL** on each target task.

| Dataset | Baseline – | Stage 1 CUB200 | Stage 2 PixmoCount | Stage 3 PathVQA | Stage 4 TextVQA | Stage 5 TimeClock |
|---|---|---|---|---|---|---|
| **Target** | | | | | | |
| CUB200 | 81.4 | 93.7 | 93.6 | 93.4 | 93.4 | 93.2 |
| PixmoCount | 58.6 | 59.9 | 53.4 | 56.7 | 53.6 | 53.4 |
| PathVQA | 29.2 | 35.7 | 35.3 | 61.0 | 57.3 | 58.3 |
| TextVQA | 83.0 | 77.8 | 77.4 | 81.8 | 83.6 | 80.6 |
| TimeClock | 8.2 | 10.5 | 9.1 | 9.6 | 9.9 | 49.4 |
| **Average** | 52.1 | 55.5 | 53.8 | 60.5 | 59.6 | 67.0 |
| **Held out** | | | | | | |
| AI2D | 82.9 | 83.3 | 82.8 | 83.3 | 82.4 | 82.2 |
| ChartQA | 83.2 | 84.4 | 79.9 | 86.8 | 86.2 | 84.5 |
| DocVQA | 94.4 | 85.6 | 92.6 | 92.9 | 93.8 | 92.7 |
| InfoVQA | 80.3 | 74.9 | 77.3 | 79.4 | 78.9 | 78.6 |
| MMStar | 62.6 | 63.6 | 63.7 | 62.4 | 61.4 | 61.8 |
| RealWorldQA | 68.6 | 70.1 | 68.1 | 67.8 | 69.2 | 68.4 |
| ScienceQA | 76.7 | 85.9 | 84.9 | 86.8 | 86.5 | 85.9 |
| SeedBench | 74.1 | 73.9 | 74.0 | 73.5 | 73.3 | 73.5 |
| **Average** | 77.9 | 77.7 | 77.9 | 79.1 | 79.0 | 78.5 |

Table 29: Detailed performance of sequentially fine-tuning the **MLP in the LLM** of **Qwen2.5-VL** on each target task.

| Dataset | Baseline – | Stage 1 CUB200 | Stage 2 PixmoCount | Stage 3 PathVQA | Stage 4 TextVQA | Stage 5 TimeClock |
|---|---|---|---|---|---|---|
| **Target** | | | | | | |
| CUB200 | 81.4 | 94.1 | 83.5 | 93.1 | 93.0 | 92.6 |
| PixmoCount | 58.6 | 56.6 | 50.0 | 53.4 | 31.8 | 52.2 |
| PathVQA | 29.2 | 32.4 | 2.8 | 61.5 | 60.6 | 60.4 |
| TextVQA | 83.0 | 76.7 | 8.2 | 78.9 | 83.4 | 64.4 |
| TimeClock | 8.2 | 7.5 | 4.8 | 6.0 | 3.3 | 60.2 |
| **Average** | 52.1 | 53.5 | 29.9 | 58.6 | 54.4 | 66.0 |
| **Held out** | | | | | | |
| AI2D | 82.9 | 78.4 | 0.1 | 81.0 | 81.0 | 74.0 |
| ChartQA | 83.2 | 83.3 | 0.0 | 74.2 | 82.2 | 74.8 |
| DocVQA | 94.4 | 82.0 | 3.0 | 89.3 | 92.5 | 72.7 |
| InfoVQA | 80.3 | 71.4 | 2.4 | 75.4 | 78.3 | 59.4 |
| MMStar | 62.6 | 60.6 | 29.0 | 58.8 | 61.4 | 55.9 |
| RealWorldQA | 68.6 | 66.1 | 4.4 | 61.4 | 65.9 | 50.7 |
| ScienceQA | 76.7 | 79.6 | 0.1 | 80.3 | 82.2 | 77.0 |
| SeedBench | 74.1 | 73.2 | 20.5 | 72.0 | 72.6 | 70.9 |
| **Average** | 77.9 | 74.3 | 7.4 | 74.0 | 77.0 | 66.9 |

Table 30: Detailed performance of sequentially fine-tuning the **MLP (Gate & Up) in the LLM** of **Qwen2.5-VL** on each target task.

| Dataset | Baseline – | Stage 1 CUB200 | Stage 2 PixmoCount | Stage 3 PathVQA | Stage 4 TextVQA | Stage 5 TimeClock |
|---|---|---|---|---|---|---|
| **Target** | | | | | | |
| CUB200 | 81.4 | 94.1 | 94.2 | 93.8 | 94.0 | 94.0 |
| PixmoCount | 58.6 | 59.7 | 49.6 | 50.0 | 50.9 | 54.7 |
| PathVQA | 29.2 | 36.0 | 22.0 | 61.5 | 61.7 | 62.3 |
| TextVQA | 83.0 | 75.9 | 74.7 | 81.1 | 83.8 | 79.6 |
| TimeClock | 8.2 | 8.1 | 6.5 | 7.6 | 5.1 | 55.5 |
| **Average** | 52.1 | 54.8 | 49.4 | 58.8 | 59.1 | 69.2 |
| **Held out** | | | | | | |
| AI2D | 82.9 | 82.3 | 76.6 | 69.0 | 80.7 | 75.5 |
| ChartQA | 83.2 | 81.8 | 81.9 | 84.1 | 76.6 | 80.5 |
| DocVQA | 94.4 | 81.9 | 81.2 | 91.3 | 93.0 | 90.4 |
| InfoVQA | 80.3 | 72.3 | 68.3 | 78.4 | 79.5 | 78.4 |
| MMStar | 62.6 | 62.5 | 47.6 | 56.0 | 59.1 | 58.7 |
| RealWorldQA | 68.6 | 68.8 | 34.5 | 56.3 | 64.4 | 66.3 |
| ScienceQA | 76.7 | 83.4 | 78.7 | 56.5 | 72.6 | 65.3 |
| SeedBench | 74.1 | 73.8 | 60.1 | 68.2 | 71.7 | 71.1 |
| **Average** | 77.9 | 75.8 | 66.1 | 70.0 | 74.7 | 73.3 |

