# OpenReview forum: "How to Teach Large Multimodal Models New Skills"
_ICLR.cc/2026/Conference — Submitted to ICLR 2026_

### Official Review · Reviewer_s6ff · 2025-10-21

**Soundness:** 3
**Presentation:** 3
**Contribution:** 3
**Rating:** 6
**Confidence:** 4

**Summary:**

This paper focuses on the continual learning problem of LMMs. Through fine-tuning experiments and a counting-bias probe, it reveals that "forgetting" of held-out tasks post-fine-tuning is essentially recoverable output token distribution shift. It further proposes two fine-tuning schemes, updating only self-attention projection layers and updating only MLP’s Gate&Up layers while freezing Down layer, which ensure strong target-task learning with minimal forgetting across multiple models and tasks.

**Strengths:**

1. Through fine-tuning experiments and a counting-bias probe, the paper first clarifies that the "forgetting" of held-out tasks in LMMs after fine-tuning is essentially a shift in the output token distribution. Moreover, this shift is partially recoverable via subsequent task fine-tuning, offering a crucial theoretical perspective on LMMs continual learning core issues.
2. The paper proposes two concise fine-tuning methods that balance strong target-task learning and minimal original-capability forgetting, validated across 3 model families, 5 target task types and 8 held-out benchmarks for robust generalization and reliability.

**Weaknesses:**

1. Regarding the question of whether the two fine-tuning methods can be combined to improve performance, Appendix F.1 notes simultaneous fine-tuning of SA Proj. and MLP (Gate&Up) offers no gain or even degrades performance, but lacks in-depth explanation of the underlying mechanism; alternative combinations (e.g., two-stage fine-tuning) are also unexplored.
2. The paper only compares its strategy with traditional methods (LoRA, WiSE-FT, MoE) and excludes recent mainstream continual learning schemes, failing to clarify the strategy’s competitiveness.
3. The "forgotten knowledge recoverability" claim lacks rigorous explanation/verification: no causal validation via experiments like "adjusting distribution without training new tasks", no quantification of distribution correction-recovery correlation, and unclear recovery triggers, reducing practical value.
4. Focused on quantitative metrics, the paper does not compare fine-tuned models’ output differences for the same input or analyze intermediate feature changes before/after forgetting recovery, hindering intuitive understanding of the strategy’s effect.

**Questions:**

1. The paper fails to quantify the training efficiency (parameter count, computation time, memory usage) of SA Proj. and MLP (Gate&Up), precluding efficiency comparison with full-model and LoRA fine-tuning. Can experimental data be supplemented to clarify applicability in resource-constrained scenarios?
2. The paper innovatively uses a counting-bias probe to verify token distribution-forgetting correlation. Can similar probes (e.g., for medical VQA, clock reading) be tested to confirm if such probes can be generalized as a universal tool for measuring task-specific token distribution shifts?

---

> ### Author Response · Authors · 2025-11-18
> **Author response**
>
> We thank Reviewer s6ff for their positive review and for recognizing our contributions regarding the "recoverable output token distribution shift" and our effective tuning schemes.
>
> We address the reviewer's weaknesses and questions below.
>
> **On combining SA Proj. and MLP (Gate&Up):**
>
> **Response:** Indeed, combining SA Proj. with MLP (Gate&Up) does not yield further benefits. Our explanation is MLP (Gate&Up) alone achieves near-maximal learning (Table 1), therefore adding SA Proj. tuning offers little room for improvement on the target task. Furthermore, because the write-back projection $W_O$ in Self-Attention acts mechanistically like a "write" operation, tuning it with the MLP (Gate&Up) has the potential to act like tuning the whole MLP. And we found that dropping $W_O$ reduces forgetting (SA Proj. (QKV) + MLP (Gate&Up)).
>
> **On comparisons with "recent mainstream continual learning schemes":**
>
> **Response:** We thank the reviewer for this suggestion. Regarding comparisons with recent state-of-the-art (SOTA) methods, we would like to clarify our position on two fronts:
>
> 1.  **Limited literature in continual learning of LMMs:** As noted in our Related Work (Section 2), the field of continual learning for general-purpose LMMs is still emerging. Most recent literature focuses on highly specialized settings (e.g., only VQA or older architectures) that are not directly applicable to our multi-skill, generative LMM setting. Therefore, we chose to compare against the four fundamental and most widely applicable categories of CL strategies: Regularization (LwF), Parameter-Efficient Tuning (LoRA), Model Expansion (MoE), and Weight Averaging (WiSE-FT).
>
> 2.  **Sufficiency of our evaluation:** Our primary goal is to investigate the *mechanism* of forgetting (output distribution shift) and identify intrinsic tuning properties of the architecture. Our extensive results demonstrate that our proposed selective tuning choices—specifically SA Proj. and MLP (Gate&Up)—are simple, effective, and achieve nearly zero forgetting on their own (Figure 4). Given that these simple recipes already solve the stability problem so effectively in this setting, we believe comparing against complex, specialized SOTA methods is not necessary to validate the core findings. Furthermore, our approach is orthogonal and can be combined with other techniques if needed, such as LwF demonstrated in the paper.
>
>  If there is any particular SOTA approach you request to compare, we are happy to check.
>
> **On the "forgotten knowledge recoverability" claim:**
>
> **Response:** The original submission already provided a rigorous explanation and direct quantification for the "recoverable forgetting" claim.
>
> Our central hypothesis, detailed in our Discovery Process (Appendix C), is that this is not "knowledge" loss but a measurable output distribution shift. We provide strong quantitative support for this in Section 5.2 (Figure 3):
> * The drop in held-out performance is shown to be **strikingly correlated** with the rise in the "number-token bias" (the output shift).
> * To make this quantification even more rigorous, we quantify the relationship between non-counting held-out performance (mid plot in Figure 3) and numeric-token likelihood on LCS-558K (right plot in Figure 3) by computing Spearman’s rank correlation across checkpoints and methods. The correlations are strongly negative (e.g., -0.84 overall), directly confirming that checkpoints with higher numeric-token likelihood systematically have worse held-out performance.
>
> While manually “adjusting the distribution without training new tasks” is non-trivial, our strong, quantified correlational evidence and clear mechanism together already provide a robust explanation for the behavior.

---

> > ### Author Response · Authors · 2025-11-18
> > **Continued author response**
> >
> > **On analyzing the forgetting/recovery process (Weakness 4):**
> >
> > **Response:** We thank the reviewer for bringing up this point. To directly address this, we have added two new analyses to the appendix, which we believe provide a comprehensive picture:
> >
> > 1.  **Intermediate Quantitative Analysis (Appendix F.6, Figure 12):** We have added a new experiment that extends our counting-bias probe to track the full forgetting-and-recovery cycle over three stages: CUB200 $\rightarrow$ PixmoCount $\rightarrow$ PathVQA. This new figure provides the direct evidence the reviewer requested. At **Stage 2 (PixmoCount)**, we see that forgetting (the drop in held-out performance) correlates directly with a **sharp spike** in the numeric output bias. Crucially, at **Stage 3 (PathVQA)**, we observe the "recovery" in held-out performance, and this correlates directly with a **significant reduction** in the numeric output bias, as the model's distribution is re-corrected by the new, more general task.
> >
> > 2.  **Qualitative Output Analysis (Appendix F.7, Figure 13):** We have also expanded our qualitative analysis to show the full cycle. This figure shows that after **Stage 2 (PixmoCount)**, the forgetting-prone models (LLM, MLP) fail on a held-out captioning task by applying their new bias (e.g., "There are 2 boots..."). Then, after **Stage 3 (PathVQA)**, these same models **regain their captioning ability** and produce the correct, descriptive caption (e.g., "A pair of grey and orange knitted boots...").
> >
> > Together, these two new analyses provide a complete, side-by-side view (both quantitative and qualitative) of the forgetting and recovery process. They confirm that "forgetting" is a temporary output distribution shift and "recovery" is the re-correction of that shift.
> >
> > **On training efficiency (parameter count, time, memory):**
> >
> > **Response:** We have added a new Section F.5 and Table 6 to the Appendix. As shown below, our methods are more efficient in throughput than LoRA and MoE.
> >
> > **Table: Training Efficiency Comparison (4x H100)**
> > | Method | Throughput (samples/sec) | Trainable Params (B) |
> > | :--- | :---: | :---: |
> > | SA Proj. | 1.46 | 0.82 |
> > | MLP (Gate&Up) | 1.45 | 3.80 |
> > | LoRA | 1.27 (Slower) | 0.50 (Smallest) |
> > | MoE | 0.44 (Slowest) | 5.70 |
> >
> > LoRA incurs overhead from adapter computations (extra forward/backward passes), and MoE requires complex routing logic. Our method uses standard dense layers, maintaining maximum training throughput.
> >
> > **On generalizing the counting-bias probe:**
> >
> > **Response:** This is an interesting question. Clock reading could be done with a similar probe, since it also has a narrowly defined response range.  Applying the same probe can be methodologically challenging for open-ended domains like medical VQA, where the set of potential "bias tokens" is vast, ambiguous, and context-dependent. Attempting to define this set *a priori* would be extremely difficult and would likely lead to a noisy, uninterpretable analysis. A more general method may be to measure the similarity of the distribution of outputs (e.g. with KL-divergence) to the initial distribution and to the target task ground truth distribution. When training, you would expect the distribution to move toward the target task and away from the initial distribution, if there is an output distribution shift.

---

### Official Review · Reviewer_xSMp · 2025-10-29

**Soundness:** 3
**Presentation:** 3
**Contribution:** 2
**Rating:** 4
**Confidence:** 3

**Summary:**

In this paper, the authors compare various layer-level fine-tuning strategies and find that selectively tuning only the self-attention projection layers or the MLP’s up and gate components allows efficient learning of new tasks while retaining performance on previous ones. Using a counting-bias probe method, the paper further shows that forgetting mainly arises from output distribution drift rather than genuine knowledge loss.

**Strengths:**

- The use of held-out benchmarks is highly valuable. Existing benchmarks typically measure forgetting only with respect to previously trained tasks within the same benchmark, without considering the preservation of the model’s intrinsic capabilities.
- The discovery that fine-tuning the Self-Attention Projection (SA Proj.) or MLP Gate&Up layers can acquire new knowledge while greatly reducing forgetting of existing abilities is both effective and practically straightforward.

**Weaknesses:**

- The conclusion that tuning SA Proj. and Gate&Up does not lead to significant forgetting has not been validated on other benchmarks. Therefore, it is difficult to rule out the possibility that this finding stems from dataset bias in the current experimental domain.
- The paper lacks direct comparisons with recent SoTA methods (published in recent two years). Although this paper demonstrates that tuning SA Proj. and Gate&Up is effective, it remains unclear how effective this approach is relative to SoTA baselines.
- The analysis of output distribution drift relies mainly on the counting-bias probe method, so it remains unclear whether the same conclusion holds for other cases where the task outputs are not primarily numeric.

**Questions:**

See weaknesses.

---

> ### Author Response · Authors · 2025-11-18
> **Author response**
>
> We thank Reviewer xSMp for their review. We are glad they found the use of held-out benchmarks "highly valuable" and our proposed methods "effective and practically straightforward."
>
> We address the reviewer's three main concerns below.
>
> **On validation and potential dataset bias:**
>
> * **Diverse Target Tasks:** We do not use a single benchmark. We test on **five practical target skills** that span different domains and answer formats: fine-grained bird classification, counting, medical VQA, OCR reading, and time reading (detailed in Appendix A).
> * **Held-out Benchmarks:** We measure forgetting not just on previously seen tasks, but on a broad suite of **eight standard, held-out benchmarks**. Importantly, several of these are **aggregated multi-task benchmarks**, for example, SeedBench reports an average over 12 distinct sub-tasks (e.g., visual reasoning, action recognition), and MMStar aggregates across 18 sub-tasks. Thus, forgetting is measured across a very broad skill set rather than a small, fixed list.
> * **Multiple Models:** We validate our findings across **three LMM model families**: LLaVA OneVision, LLaVA-NeXT, and Qwen2.5-VL.
>
> The fact that our key findings hold true across this diverse set of tasks, aggregated benchmarks, and models supports the generality of our conclusions.
>
> **On comparison with SOTA methods:**
>
> **Response:** We thank the reviewer for this suggestion. Regarding comparisons with recent state-of-the-art (SOTA) methods, we would like to clarify our position on two fronts:
>
> 1.  **Limited literature in continual learning of LMMs:** As noted in our Related Work (Section 2), the field of continual learning for general-purpose LMMs is still emerging. Most recent literature focuses on highly specialized settings (e.g., only VQA or older architectures) that are not directly applicable to our multi-skill, generative LMM setting. Therefore, we chose to compare against the four fundamental and most widely applicable categories of CL strategies: Regularization (LwF), Parameter-Efficient Tuning (LoRA), Model Expansion (MoE), and Weight Averaging (WiSE-FT).
>
> 2.  **Sufficiency of our evaluation:** Our primary goal is to investigate the *mechanism* of forgetting (output distribution shift) and identify intrinsic tuning properties of the architecture. Our extensive results demonstrate that our proposed selective tuning choices—specifically SA Proj. and MLP (Gate&Up)—are simple, effective, and achieve nearly zero forgetting on their own (Figure 4). Given that these simple recipes already solve the stability problem so effectively in this setting, we believe comparing against complex, specialized SOTA methods is not necessary to validate the core findings. Furthermore, our approach is orthogonal and can be combined with other techniques if needed such as LwF demonstrated in the paper.
>
> If there is any particular SOTA approach you request to compare, we are happy to check.
>
> **On the generality of the output distribution drift analysis:**
>
> **Response:** Thank you for the question. We used the "counting-bias probe" as our clearest *quantitative* illustration for a specific methodological reason: its "bias set" (numeric tokens) is small, unambiguous, and clearly identifiable in the vocabulary. This allows for a clean, low-noise measurement. In contrast, defining a similar "bias set" for a task like Medical VQA is methodologically difficult, as the 'medical tokens' are vast and context-dependent.
>
> Our conclusion that this is a *general* phenomenon is supported by two other key observations that do not rely on a numeric probe:
>
> 1.  **Recoverable Forgetting as a General Probe:** Our strongest evidence is the "recoverable" nature of forgetting, shown in Figure 1 (Right) and Appendix E.2 (Figs. 5-7). Performance drops sharply after the *narrow-output* counting task but then **recovers** when the model is trained on the *diverse-output* PathVQA task. This "drop and rebound" acts as a general, non-numeric probe. It strongly suggests the "forgetting" was a temporary **output bias** that was simply "re-corrected" by the subsequent, more general task, rather than a permanent loss of concepts.
>
> 2.  **Qualitative Illustration of the Bias:** We also provide a direct illustration of this bias in Appendix F.4 (Figure 11). On a **held-out image captioning task**, the MLP-tuned model fails by applying its new bias (answering "There are 2 photos in the photo."). This qualitatively shows the failure is a shift in *output format*, not a loss of *underlying concepts* (it still correctly identifies "photos").
>
> Thus, the counting probe (Fig. 3) quantifies the mechanism in a clean setting, while the recovery phenomenon (Fig. 1) and qualitative results (Fig. 11) demonstrate its generality.

---

### Official Review · Reviewer_hD4Q · 2025-10-29

**Soundness:** 2
**Presentation:** 3
**Contribution:** 2
**Rating:** 4
**Confidence:** 3

**Summary:**

This paper studies continual learning and forgetting in VLM training. The authors propose two simple but effective methods to mitigate the forgetting, that are, (1) only updating the attention and projection (Wq, Wk, Wv, Wo), and (2) updating only the MLP gate and up layers. run experiments on different tasks. The authors do test on 5 different tasks sequence, 8 held-out benchmarks, and 3 different VLM families to verify the effectiveness of the simple methods. There is also an interesting understanding part, trying to map the forgetting to the token-distribution shift.

**Strengths:**

(1) very clear writing: from my side, it is easy to understand
(2) easy but effective method
(3) experiments are relatively comprehensive on my side
(4) i really appreciate the understanding part, where the authors dive deeper into the reason of forgetting, and map it into the token-distribution shift.

**Weaknesses:**

The main contribution of the paper is to study which part of the parameters to update (to my understanding, correct me if I am wrong). While indeed the proposed methods already show signal, it is not clear if there is any logic/reasons behind selecting those parameters. There are many other confounding factors, which may make the conclusion change. for example
(1) If the model is larger, are there any other rules for selecting the update parameters?
(2) If the model is larger, will it be beneficial to use LoRA?
(3) Will it be better to select the parameters based on the layer index, e.g., if it is better to update on a later layer than an earlier layer?
I am afraid that in the end, it will just turn into an engineering problem, where you just run all the design choices and pick the best, it there are limited scientific guides.

However, I still agree that the token distribution is interesting.

**Questions:**

Please see the weakness part. my main questions are regarding these confounding factors.

---

> ### Author Response · Authors · 2025-11-18
> **Author response**
>
> We thank Reviewer hD4Q for their valuable feedback. We are very encouraged that they found the writing "very clear," the method "easy but effective," and the experiments "comprehensive."
>
> We address the reviewer's main concerns regarding the logic and generalizability of our findings.
>
> **On the logic behind selecting SA Proj. and MLP (Gate&Up):**
>
> **Response:** We respectfully disagree that the choice of parameters does not have clear logic/reasons. Our selection is directly motivated by a mechanistic hypothesis derived from established transformer literature (Section 2) and our own discovery process (Appendix C).
>
> Prior work (e.g., Geva et al., 2021; Olsson et al., 2022) establishes that Self-Attention (SA) layers primarily act as "routers" (processing inputs), while MLPs function as "key-value memories" that "write" features to the output distribution. Our methods (Sec 3.2) are designed to isolate these roles: SA Proj. tuning isolates routing, while MLP (Gate&Up) tuning isolates concept activation ($W_{\text{gate}}, W_{\text{up}}$) while freezing the *write-back* mechanism ($W_{\text{down}}$). The fact that these specific configurations succeed (Table 1) where full tuning fails **confirms** the hypothesis that unregulated "writing" to the residual stream is the primary driver of forgetting.
>
> **On confounding factors (model size, LoRA, layer index):**
>
> **Response:**
>
> 1.  **Model Size:** We validated our results across **three different model families** (LLaVA-OneVision, LLaVA-NeXT, Qwen2.5-VL) with broad benchmarks. While testing on even larger models would be interesting, it is prohibitively expensive for us, as noted in our limitations (Section 6) and the Discovery Process (Appendix C). Not doing these experiments should not count against us.
>
> 2.  **LoRA on larger models:** Thanks for this interesting question. As shown in the added Section F.5 and Table 6, LoRA is more efficient in parameters but not directly more efficient in training. While we agree that LoRA can be more parameter efficient in larger models, we found LoRA's effectiveness to be highly sensitive to hyperparameters, which is a critical drawback in a *sequential* learning setting where re-tuning for every new task is impractical. This finding aligns with other work, such as Biderman et al. (2024), who also note LoRA's sensitivity to hyperparameters like the learning rate in their appendix.
>
>     Furthermore, as shown in the table below (derived from Appendix Tables 6, 8, and 15), LoRA is actually **slower** in training throughput than our method and performs worse on both learning and forgetting:
>
>     **Table: Efficiency & Stability Comparison (LLaVA-OneVision)**
>
>     | Method | Training Speed (samples/sec) | Target Learning | Held-out Forgetting |
>     | :--- | :---: | :---: | :---: |
>     | LoRA | 1.27 (Slower) | +20.0% | -4.2% |
>     | **SA Proj.** | **1.46 (Faster)** | **+24.3%** | **+0.1%** |
>
>     Therefore, while LoRA offers parameter efficiency, its instability and adapter overhead pose challenges for dynamic, continual learning streams.
>
> 3.  **Layer Index:** We performed an analysis of output distribution shift (Appendix F.2, Figure 9), finding that the shift occurs overwhelmingly in late layers. In response to your question, we also experimented with tuning only the first 23 or last 5 layers and found that tuning only the first 23 layers gives the full benefit on target tasks with less forgetting (new Appendix F.3, Fig 10).
> However, we want to emphasize that our method is not to tune individual layers, but to limit tuning to the SA Proj or MLP up components of all layers. While choosing individual layers to tune could save memory or improve performance in some cases, we agree that it adds a lot of experimental complexity, and thus we propose a simpler solution that consistently works for reasons that we explain, while leaving open the possibility of further tweaking.

---

### Official Review · Reviewer_yr6z · 2025-10-31

**Soundness:** 3
**Presentation:** 3
**Contribution:** 2
**Rating:** 6
**Confidence:** 4

**Summary:**

The paper studies effective continual learning strategies in multimodal models. Major findings are: (a) Updating a selected set of parameters can reduce forgetting, and (b) forgetting can be traced in distribution shift in output tokens. The authors corroborate their study with extensive experiments on five target skills while monitoring general ability on eight held‑out benchmarks across three model families.

**Strengths:**

The main strength of the paper lies in its clear and simplistic presentation of the continual training experiments. By varying the parameters for updating in this sequence learning setting, they clearly show that selecting the right parameters to update can substantially reduce forgetting. Furthermore, through a mechanistic analysis, they connect forgetting behavior to mechanistic roles of attention vs. MLP layers. Through extensive experiments on wide variety of benchmarks and backbone models, the authors show that forgetting can be mitigated with a simple fine-tuning recipe.

**Weaknesses:**

As such, I don't find any key weaknesses with the paper. I have some questions about the experiment setup:

a) All tasks are vision-language. Do the findings generalize if a text-only task was included in the sequence? If not, is it primarily the issue of the way the backbone LLM has been converted to the multimodal LLM?

b) How do the results change if you vary the order of the tasks in sequence? The authors evaluate multiple task orders but do not report variance or confidence intervals (e.g. in table 1).

**Questions:**

Please see above for my questions.

---

> ### Author Response · Authors · 2025-11-18
> **Author response**
>
> We thank Reviewer yr6z for their constructive feedback. We are glad they found the presentation "clear and simplistic" and appreciated the "mechanistic analysis" connecting forgetting to output distribution shift.
>
> We address the reviewer's two main questions below.
>
> **On generalizing to text-only tasks:**
>
> **Response:** This is an excellent question. Our study's focus is on Large Multimodal Models (LMMs), so our experiments were designed to test the addition of new *vision-language* skills (like counting, time reading, etc.), as described in Section 4.1 and Appendix A.
>
> However, our core findings and the mechanism we identify are rooted in the **transformer decoder (LLM) component**. Our analysis distinguishes between the mechanistic roles of self-attention (SA) as a routing mechanism and MLPs as memory layers that "write" to the output distribution (as discussed in our Introduction and Related Work sections). Since this is a fundamental property of the transformer architecture itself, we hypothesize that these findings are highly likely to generalize to continual learning in text-only LLMs. We agree that rigorously testing this is an important direction for future work.
>
> **On the effect of task order and variance:**
>
> **Response:** This is an important point. As the reviewer noted, the metrics in **Table 1** are averaged over three distinct five-task curricula to ensure robustness.
>
> To explicitly address the variance:
> 1.  **Full Trajectories:** We provided the full sequential performance curves for all three task orders in Appendix E.2 (Figures 5, 6, and 7). These plots demonstrate that our key findings, such as the stability of SA Proj. and the recovery of forgetting, are consistent across different permutations.
> 2.  **Statistical Significance:** We have updated Table 1 to include statistical significance testing. As described in the revised Section 5.1 (modifications are marked with blue text), we run paired sample t-tests on the per-sequence/per-task averages. In the updated table, we underline results that are *not* significantly different ($p > 0.1$) from the best method. As summarized below, our proposed methods provide statistically significant stability compared to Full Fine-tuning.
>
> **Table: Statistical Significance of Stability (vs. Best Method)**
> | Method | Avg. Held-Out Drop | Significant Difference from Best? |
> | :--- | :---: | :---: |
> | Full Finetuning | -27.4% | **Yes** (Significantly worse) |
> | LLM Only | -23.3% | **Yes** (Significantly worse) |
> | **SA Proj.** | **-0.6%** | **No** (Statistically comparable to best) |
> | **MLP (Gate&Up)** | **-2.1%** | **No** (Statistically comparable to best) |
>
> This analysis confirms that the superior stability of our proposed methods is statistically significant across the varied orders.

---

### Author Response · Authors · 2025-11-26

To not lose the forest through the trees, we want to point out that the reviewers agree that the paper, with clarity of writing and extensive experiments, improves our understanding that much "forgetting" observed when learning new skills is actually shift of output distribution, and the paper offers a simple, practical, effective solution to mitigate this forgetting. We believe this is a valuable and interesting contribution, which has been further strengthened by revision in response to excellent reviewer comments and questions.

---

### Author Response · Authors · 2025-11-30
**Summary of Reviewer Feedback and Rebuttal Clarifications**

Dear AC, we are sorry to hear about the OpenReview leak. To reduce the strain on you and help our paper be fairly evaluated, we have summarized the reviews and responses below with a metareview (using GPT to avoid slanting). We tried to be objective in summarizing strenghts, weaknesses, and responses, and put our opinions in notes to AC.

**Metareview**

**Summary** (GPT summary of reviewer summaries)
The paper studies effective continual learning strategies and forgetting in multimodal models. It compares various layer-level fine-tuning strategies and finds that selectively tuning "only the self-attention projection layers (Wq, Wk, Wv, Wo)" or "only the MLP's Gate&Up layers while freezing Down" enables efficient learning of new tasks with "minimal forgetting." The authors run "extensive experiments on five target skills" in a 5-task sequence, while monitoring general ability on "eight held-out benchmarks" across "three different VLM families." Using a counting-bias probe, they show that apparent forgetting is "essentially recoverable output token distribution shift" — i.e., "output distribution drift rather than genuine knowledge loss"—thus tracing forgetting to "distribution shift in output tokens."

**Strengths and Weaknesses** (GPT summary of notes below)

Across reviews R1–R4, the paper's **strengths** are consistent: clear writing and experimental presentation, simple yet effective fine-tuning schemes (tuning only self-attention projections or MLP Gate&Up), extensive experiments (5 target skills, 8 held-out benchmarks, 3 model families), and a mechanistic analysis that links forgetting to output token distribution shift using held-out benchmarks and a counting-bias probe.

**Weaknesses** center on: (1) questions about experimental setup (R1 yr6x), (2) how generally one can decide which parameters to tune and whether that is model-dependent (R2 hD4Q), (3) whether conclusions hold beyond the reported benchmarks and relative to recent SoTA continual-learning methods (R3 xSMp, R4 s6ff), (4) reliance on the counting-bias probe and lack of more intuitive qualitative comparisons (R3, R4), and (5) the unexplained degradation when jointly tuning SA Proj + MLP (R4).

The author **responses** directly tackle these points: they provide additional statistical tests, highlight existing interventional and mechanistic analyses that motivate the chosen parameter subsets, and emphasize the breadth of benchmarks and model families already covered (which other reviewers praised). They argue that comparable LMM continual-learning baselines are scarce and that they already evaluate a reasonable range of mitigation methods. They add per-layer analyses, further quantitative evidence (e.g., Spearman 0.84 between distribution shift and forgetting) and new qualitative examples, plus an explanation of the joint-tuning degradation.

**Overall**, the concerns are largely addressed, and the authors note that the textual comments of multiple reviewers (especially R1 and R2, and even R4’s strengths) read as a clear accept, making the "marginal accept" scores appear inconsistent with the stated strengths and the rebuttal.

## --- Condensed reviews and responses ---

**Paper summary** (quoted from R4):
This paper focuses on the continual learning problem of LMMs. Through fine-tuning experiments and a counting-bias probe, it reveals that "forgetting" of held-out tasks post-fine-tuning is essentially recoverable output token distribution shift. It further proposes two fine-tuning schemes, updating only self-attention projection layers and updating only MLP's Gate&Up layers while freezing Down layer, which ensure strong target-task learning with minimal forgetting across multiple models and tasks.

**R1 (yr6x)**

Strengths
* clear presentation of continual learning experiments
* clearly show that tuning self-attention projection or MLP up layers substantially reduces forgetting
* mechanistic analysis connects forgetting behavior to roles of attention and MLP layers
* extensive experiments on wide variety of benchmarks and backbone models

Weaknesses
1. no key weaknesses, but two questions about the experimental setup

Response
1.  addresses questions based on supplemental material with additional statistical testing

Note to AC
* The reviewer's comments in summary and strengths and weaknesses look like a clear accept.

---

> ### Author Response · Authors · 2025-11-30
> **Continued Summary of Reviewer Feedback and Rebuttal Clarifications**
>
> **R2 (hD4Q)**
>
> Strengths
> * clear writing
> * simple and effective method
> * comprehensive experiments
> * appreciate that paper uncovers the reason for forgetting and maps it to token distribution shift
>
> Weaknesses
> 1. not clear if there is logic behind which parameters to tune and how much it depends on the type of model; could require engineering for each case
>
> Response
>
> **1a**. points to sections of the paper where we support which parameters to tune based on interventive experiments, mechanistic analysis, and corroborative related work
> **1b**. points to experiments in the paper with varying model families and LoRA
> **1c**. adds new per-layer analysis (some was already in supplemental)
>
> Note to AC
> * As indicated by other reviewers, the submitted paper already addresses the main concern directly with extensive experiments, mechanistic analysis, explanation of output shift, and corroborating recent work. Given that and the listed strengths, the authors believe this should be a positive rating.
>
> **R3 (xSMp)**
>
> Strengths
> * use of held-out benchmarks is highly valuable
> * discovery that fine-tuning the Self-Attention Projection (SA Proj.) or MLP Gate&Up layers can acquire new knowledge while greatly reducing forgetting of existing abilities is both effective and practically straightforward
>
> Weaknesses
> 1. conclusion that tuning SA Proj. and Gate&Up does not lead to significant forgetting has not been validated on other benchmarks
> 2. paper lacks direct comparisons with recent SoTA methods (no specific reference indicated by reviewer)
> 3. analysis of output distribution drift relies mainly on the counting-bias probe method
>
> Response
> 1. points out that the evaluation includes five target skills, eight multitask benchmarks encompassing dozens of tasks, and three model families (also see R1 and R4 strengths)
> 2. as noted in related work section, few works exist to compare in LMM tuning, and we did not find any that experiment in the kind of general multi-task scenario that we are exploring, so we evaluate a broad range of forgetting mitigation methods that one could reasonably try
> 3. points to other supporting evidence for conclusions about output distribution shift in the paper and adds qualitative results to support
>
> Note to AC
> * The main concern about experimental support is contradicted by strengths listed by other reviewers, and the authors believe all concerns are addressed in the rebuttal.
>
> **R4 (s6ff)**
>
> Strengths
> * through fine-tuning experiments and a counting-bias probe, the paper clarifies that the "forgetting" of held-out tasks in LMMs after fine-tuning is essentially a shift in the output token distribution
> * shows that shift is partially recoverable via subsequent task fine-tuning, offering a crucial theoretical perspective on LMMs continual learning core issues
> * proposes two concise fine-tuning methods that balance strong target-task learning and minimal original-capability forgetting
> * validated across 3 model families, 5 target task types and 8 held-out benchmarks for robust generalization and reliability
>
> Weaknesses
> 1. appendix shows that jointly tuning SA Proj and MLP up can degrade performance, which is not well explained, and some other combinations are not tested
> 2. paper only compares with traditional methods (LoRA, WiSE-FT, MoE), not recent mainstream continual learning schemes (no reference provided by reviewer)
> 3. "forgotten knowledge recoverability" claim lacks rigorous explanation/verification
> 4. paper focuses on quantitative metrics and does not supply sufficient intuitive understanding by comparing outputs for the same input from different models
>
> Response
> 1. provides a brief explanation of this result
> 2. similar response to R3 regarding related work and comparisons
> 3. points to Appendix C, which summarizes the discovery process and explains the strong evidence for this conclusion; adds Spearman's rank correlation (0.84) to quantify strong relationship between output distribution shift and measured forgetting
> 4. additional quantitative and qualitative evidence added to supplemental
>
> Note to AC
> * The strengths noted by this reviewer are substantial, and the authors feel these concerns have been fully addressed
>
> Overall note to AC: The authors felt that the comments are consistently more favorable to the paper than the ratings. Out of curiosity, we ask GPT to guess the each reviewer rating based only the reviewer's comments, and it guessed 7, 6, 6, 7.

---

### Meta-Review · Area_Chair_Ppcv · 2026-01-07

**Summary:**

The reviewers found this paper to be clearly written, well-motivated, and thoroughly experimental, employing a simple yet effective fine-tuning strategy that mitigates forgetting in continual multimodal learning. The experiments across multiple backbones, task sequences, and held-out benchmarks is considered a major strength, and the mechanistic analysis linking forgetting to output token distribution shift is regarded as insightful and valuable. However, reviewers raise several limitations: the conclusions may be biased by the vision-languag (only setting and limited benchmark diversity); variance across task orders is not fully quantified; and the rationale for selecting specific parameter subsets to update remains underexplained, raising concerns that the approach may be largely empirical or engineering-driven. In addition, comparisons with recent state-of-the-art continual learning methods are insufficient. Some analytical statements lack rigorous verification or broader exploration, such as the absence of universal applicability in explanations of forgettable recoverability and distribution transfer. The paper is seen as a empirical and analytical contribution, but one that would benefit from clearer theoretical grounding, stronger SoTA comparisons, and broader validation of its key findings. Therefore, AC's recommendation is to reject.

**Reviewer Concerns:**

The authors quantify the differences in task sequences in their rebuttal and explain the potential for generalizing the summarized patterns to pure text tasks. Given that the experimental benchmarks employed in this paper encompass a variety of multimodal tasks designed to evaluate different capabilities of LMMs, concerns regarding benchmark diversity can be addressed. The fundamental rationale for selecting specific parameter subsets for updating has not been explicitly summarized. The rebuttal merely explains the roles played by different parameters. This significantly undermines the generalizability of the proposed view, especially considering that no experimental verification has been conducted on larger-scale models. Additionally, several reviewers mentioned comparisons with the most recent state-of-the-art continuous learning methods, which were not directly addressed.

**Reviewer Scores:**

I expect the final rating to be as follows:
- Reviewer yr6z: 6
- Reviewer hD4Q: 4
- Reviewer xSMp: 4
- Reviewer s6ff: 6

---

### Decision · Program_Chairs · 2026-01-26

Reject